



**Spatiotemporal changes of drought area as input for a machine-learning**
**approach for crop yield prediction**
Vitali Diaz[1,2], Ahmed A.A. Osman[3], Gerald A. Corzo Perez[1,2], Henny A.J. Van Lanen[4],
Shreedhar Maskey[1], Dimitri Solomatine[1,2,5]
[1]IHE Delft Institute for Water Education, Delft, 2601 DA, the Netherlands
[2]Delft University of Technology, Delft, the Netherlands
[3]Arcadis, Wales, United Kingdom
[4]Hydrology and Quantitative Water Management Group, Wageningen University, Wageningen, the Netherlands
[5]Water Problems Institute of the Russian Academy of Sciences, Moscow, Russia
**Corresponding author**: Vitali Diaz; v.diazmercado@tudelft.nl; vitalidime@gmail.com
**Abstract**
Climate change has increased the possibility of more severe and prolonged droughts worldwide, which
requires innovative methods to predict their impacts on different sectors such as agriculture. Crop
growth models calculate yield and variables related to plant development and are used for crop yield
estimation, a useful variable for monitoring drought impacts. Although used for prediction, these crop
models are not explicit forecasting models; they are limited to the physical assumptions reflected in
their conceptual model. In addition, the input data availability, the spatial and temporal aggregation,
and different sources of uncertainty make the crop yield prediction challenging. Given these limitations,
machine learning (ML) models are often utilised following a multivariable forecasting approach, but
their use with the spatial characteristics of droughts as input data is limited. This research explored the
spatial extent of drought as input data for building an approach for predicting seasonal crop yield. This
ML approach is made up of two components. The first includes polynomial regression (PR) models,
and the second considers artificial neural network (ANN) models. This approach aimed to evaluate both
types of ML models (PR and ANN) and integrate them into one operational tool. The logic is as follows:
ANN models determine the most accurate predictions, but in practice, issues regarding data retrieval
and processing can make the use of equations, i.e. PR, preferable. The proposed approach provides
these PR equations with early and preliminary input to perform such calculations. The estimates can be
further improved when the ANN models are run with the final input data. The results indicated that the
empirical equations (PR) produced good predictions when using drought area as the input. ANN
provides better estimates, in general. The results presented are a proof of concept showing the
capabilities of this ML approach to predict drought impacts with a certain degree of confidence.
Research results show that the spatiotemporal changes of drought area and its temporal aggregation
provide an important pre-processing alternative to implement  ML models for drought impact
prediction.
**Keywords**
Spatio-temporal analysis, crop yield, drought impact, machine learning, agricultural drought



## 1 Introduction

Drought frequently hits many regions across the world. It negatively affects various human activities such as agriculture, which not only generates economic losses but can also trigger famine, causing millions of deaths (Below et al., 2007; Food and Agriculture Organization of the United Nations (FAO), 2017; Kim et al., 2019; Sheffield and Wood, 2011; World Meteorological Organization (WMO), 2006). Hence, methods that help to improve strategies for drought mitigation are necessary. Within these methods are those that allow predicting the impacts of drought.

Assessments of drought impacts confirm that the presence of drought on human activities can be devastating. For instance, the Food and Agriculture Organization of the United Nations (FAO) calculated the damage and losses in the agricultural sector caused by five types of hazards, including drought. FAO estimates that drought causes damages and losses to the agricultural sector by up to 80% (FAO, 2017). Although multiple factors are involved in agriculture affectation, drought often plays the primary role, as literature confirms (Dai, 2011; FAO, 2017; Kim et al., 2019).

The assessment of drought impacts on agriculture can be performed with the help of crop yield. FAO defines crop yield as the measure of the yield of a crop per unit area of land cultivation (in kg/ha or ton/ha) (FAO and DWFI, 2015). For assessing crop yield under drought affectation, physical models based on crop properties turn out to be more comprehensive and descriptive (Huang et al., 2019; Reynolds et al., 2000; White et al., 1997; Wu et al., 2016). However, an important barrier to such models' realisation is the lack of detailed crop data and the difficulty representing all the processes involved in all stages of crop development (Huang et al., 2019; Reynolds et al., 2000; Wu et al., 2016).

Statistical and machine-learning (ML) models, which involve mathematical equations to calculate the output of a model with suitable input(s), can be used to assess crop yield impact by drought without considering any biological or physical process of the crop but the analysis of the input and output data (Araneda-Cabrera et al., 2021; Chlingaryan et al., 2018; Rahmati et al., 2020; Udmale et al., 2020; van Klompenburg et al., 2020). There have been studies where various inputs, ML techniques, and architectures (configurations) have been tested for crop yield prediction mainly following a multivariable forecasting approach (e.g., Chlingaryan et al., 2018; van Klompenburg et al., 2020). However, the use of spatial characteristics of drought such as its spatial extent has not been fully explored to crop yield prediction. The prediction refers to the calculation of crop yield at the end of the growing season (harvesting) with




information available before or during the crop development season (pre-harvesting). Previous
studies have found the spatial extent of drought to be highly correlated with the variation of
crop yield, which motivates its use in the construction of crop yield prediction models in this
research (Araneda-Cabrera et al., 2021; Diaz et al., 2016; Osman, 2018; Osman et al., 2018).
This research aims to develop an ML approach to calculate seasonal crop yield (CY) with the
monthly drought areas (DAs) as input. The ML approach comprises two components. Each
component includes a set of the following types of ML models: polynomial regression (PR)
and artificial neural network (ANN). The goal is to build both types of ML models (ANN and
MR) and use them as an integrated tool to support the decisions made based on crop yield
prediction. The logic is as follows. PR provides the prediction where the crop yield calculation
is "clear" to the performer (the end-user) because she/he has access to the equations that have
a straightforward interpretation and calculations can be done with early and preliminary input
data. For its part, ANN is used as the most accurate model, although the output calculation is
not as "clear" as in the case of PR due to the difficulty of interpreting the structure of the
resulting ANN. The ANN is expected to be used with the final input data.
Three East Indian regions where agriculture plays an important role were chosen as a case
study. ML models were built for the period 1967-2015. ML models aim to predict rice crop
yield since rice is the most cultivated crop in these regions. The ML approach was applied
separately in each of the three regions.
**Crop yield prediction in India**
In India, as in many other countries, the official crop yield prediction is mainly based on
conventional data collections techniques such as ground-field visits (Bhatt et al., 2014;
Reynolds et al., 2000; Sawasawa, 2003). The crop yield is measured through crop cutting
experiments carried out over sample crop areas. In this country, crops' area and yield
calculations are released through the Directorate of Economics and Statistics, Ministry of
Agriculture (DESMOA). A specific crop's production (in kg or ton) is calculated by
multiplying the whole field area (cultivation district) by its crop yield. The crop production is
needed for the decision-makers to take various policy decisions relating to pricing, marketing,
distribution, exportation and importation.
The Kharif season, as it is locally known, represents about 80% of the annual rainfall (Naresh
Kumar et al., 2012). This monsoon season generally goes from June to October. In this season,
the highest agricultural production is obtained. Estimation of Kharif crop yield and production
is released four times during the year with different levels of sophistication and precision,



where the last one is considered the most accurate. The first calculation is presented in
September, the second one in January, the third one in March/April, and the fourth, and the last
one, in June/July. It should be noted that the last two calculations of crop yield and production
become available much after the crops have already been harvested in December/January.
From the four calculations, the first two can be considered predictions. These two first
predictions serve as primary estimations about how much the final yield and production will
be.
The existing ground-field visits-based crop yield calculation system provides reliable
information for various crops, including rice, at the district, state, and country level for each of
the four realisations previously described; however, it lacks pre-harvesting forecasting. This
limitation of crop yield prediction motivated the creation of a satellite-based forecasting system
to have information at the early stages of crop growth. The system is carried out by the
Mahalanobis National Crop Forecast Centre (NCFC) (Sawasawa, 2003). The NCFC system is
continuously verified and updated. Although the NCFC system advances the one based on
ground-field visits by providing information in the early stages of crop growth, the data
required for its execution may not always be available. Therefore, it is necessary to explore
other solutions.
In this study, it is not intended to replace the previous and new forecasting systems in India but
to provide a complement to corroborate calculations from both types of systems and, in a
broader sense, to provide the scientific community with an approach to crop yield prediction
with information on the spatial extent of drought.
**2 Data**
**2.1 Crop yield**
Rice is the most important food grain in East India, so it was selected to assess our ML-oriented
crop-yield predictions. Rice from this region accounts for roughly 85 percent of the total rice
production in India (Ghosh et al., 2014). As mentioned, ML models were constructed for three
regions of the eastern Indian (Figure 1). State-wise crop-yield data was retrieved from 1966 to
2015 (49 years) through the Indian Directorate of Economic and Statistics from the Department
of Agriculture (DAC) (http://eands.dacnet.nic.in/).
Time series of crop yield data were arranged as follows. There are three crop seasons in India:
Rabi, Kharif, and Zaid. Of these, the Kharif season was chosen for study because it is the largest
in terms of crop production. Kharif crops are sown in June and harvested in
November/December. Seasonal crop-yield data was obtained from the DAC website and


arranged into time series per region. In this way, one value was assigned to each year of crops
harvested in the Kharif season (Figure 1). In the arrangement of the time series of the yield
data, no data filling was carried out since there are data for each year in the three regions.
Figure 1 also shows the location of the three regions. These regions are made up as follows.
Region 1 includes the current states of Bihar and Jharkhand; region 2 corresponds to the state
of West Bengal; and region 3 makes up the state of Odisha. Two important clarifications have
to be made regarding crop yield data retrieving for these regions. First, in late 2000, Bihar was
divided into two states: Bihar and Jharkhand. Thereafter, rice data was reported separately. In
this study, both states are marked as region 1; the crop-yield data from 2000 to 2015 is the
reported sum of current Bihar and Jharkhand. Second, in 2011, Orissa was renamed Odisha
(region 3), but the territory remains the same. In this case, crop yield data for Odisha is that
reported for the former Orissa and the current Odisha.

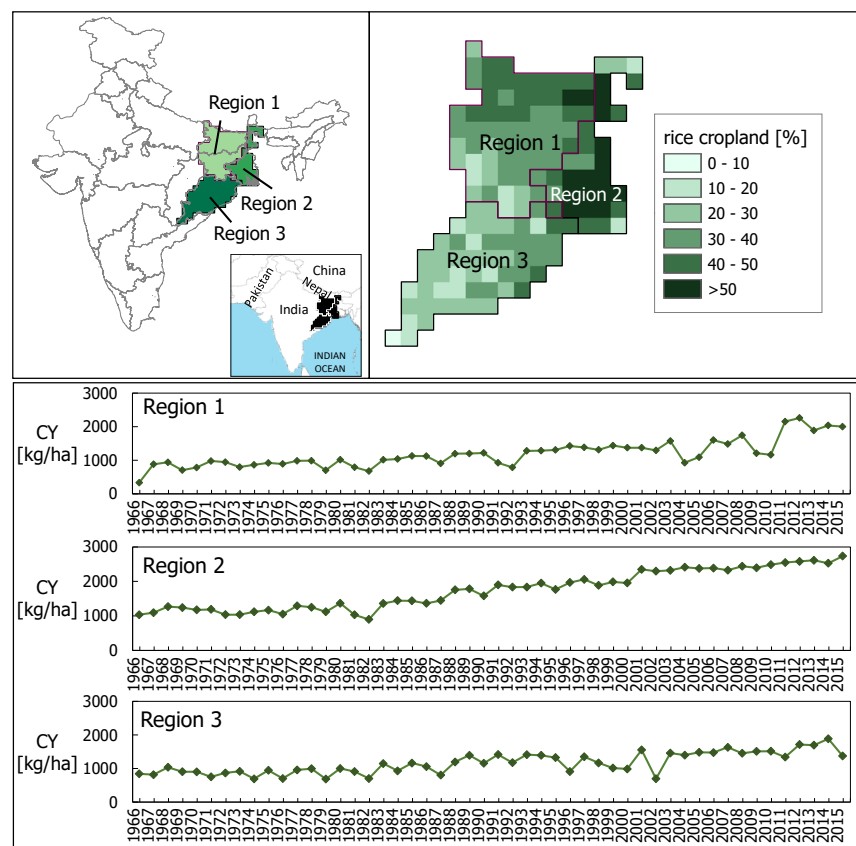


**Figure 1** Case study location (top) and crop yield (CY) data (bottom). Case study comprises region 1 (Bihar and
Jharkhand), region 2 (West Bengal), and region 3 (Odisha). The rice cropland (in percentage) is indicated. Source
of rice cropland: Monfreda et al. (2008).


## 2.2 Drought indicator

Soil moisture is the preferred variable for calculating agricultural drought indicators. However, another widely disseminated way to indirectly infer this type of drought indicator is to use meteorological drought indicators as proxies. Among these, the Standardised Precipitation Evaporation Index (SPEI) proposed by Vicente-Serrano et al. (2010) has shown to be useful in assessing agricultural drought. The SPEI follows a similar methodology as that of the widely used Standardized Precipitation Index (SPI) (Mckee et al., 1993), but with added consideration for the difference between precipitation and evapotranspiration. SPEI data was retrieved from the SPEI Global Drought Monitor (https://spei.csic.es) between 1901 and 2015. The spatial resolution of the drought indicator data is 0.5 degrees. The SPEI data was available at different aggregation periods; for this study, it was retrieved for the aggregation periods of 1, 3, 6, 9, and 12 months, indicated as DI1, DI3, DI6, DI9, and DI12, respectively.

## 3 ML modelling methodology

The experiment was carried out with the following methodology that involves the ML construction. The next paragraphs show each step in detail. These steps are (1) data preparation, (2) input variable selection, (3) polynomial regression models calculation, (4) artificial neural network models calculation, and (5) models application and combination.

### 3.1 Step 1. Data preparation

Two types of data were prepared, the crop yield (CY) and the drought areas (DA). For data preparation, three tasks were carried out (1) data retrieving, (2) drought areas calculation, and (3) data de-trending.

### 3.1.1 Data retrieving

Section 2 showed what corresponds to data retrieving for crop yield (CY) and the drought indicator (DI). A summary of CY and DI is as follows. Seasonal CY data correspond to the largest growing season. CY time series has a value for each year for the period 1966-2015 (49 years). CY was available for each region. On the other hand, drought indicator data is on a monthly basis for the period 1901-2015. The spatial resolution is half a degree.

### 3.1.2 Drought areas calculation

The drought areas were calculated following the methodology presented below. These areas were calculated for the three regions. Drought areas were calculated from the drought indicator data that is in a grid format, i.e., each cell has associated a geographic location and a time step. The calculation of drought areas started with the reclassification of all the cells of the drought indicator data by non-drought and drought cells. The drought indicator data was evaluated cell

by cell to determine those that are in drought, i.e. drought condition. To determine drought and
non-drought condition ($D_S$), the Eq. 1 was applied (Corzo Perez et al., 2011; Diaz et al., 2019,
2020; Herrera-Estrada et al., 2017). Eq. 1 represents the following. When the drought indicator
is below to the chosen threshold $\tau$, the value of 1 is used to indicate drought in the cell and non-
drought is represented by the value of 0. This classification is performed for all the cells of the
grid data in each time step ($t$).
$$D_S(t) = \begin{cases} 1 & \text{if } \mathrm{DI}(t) \leq \tau \\ 0 & \text{if } \mathrm{DI}(t) > \tau \end{cases} \qquad \text{(Eq. 1)}$$
Once the ones-and-zeros data was obtained, the drought areas (DAs) were calculated for each
region with Eq. 2. DA was computed as the ratio between the cells in drought and the total
number of cells of the region ($N$). In Eq. 2, the number of cell is denoted by $c$.
$$\mathrm{DA}(t) = 100/N \cdot \sum_{c=1}^{N} D_S(t) \qquad \text{(Eq. 2)}$$
The number of cells ($N$) of the mask is 63, 31 and 54 for region 1, 2 and 3. The masks in raster
format were built for each region. The mask is an array of ones and zeros, where the value of
1 indicates the land. We used the threshold $\tau = -1$ to calculate cells in droughts. This threshold
is widely used to identify a cell in drought when working with standardised indices such as the
used in this research (Sect. 2.2). Usually, drought indicator data is calculated at different
aggregations periods. We retrieved this data for 1, 3, 6, 9, and 12 months of aggregation period
(Sect. 2.2).  DAs' time series were calculated for each aggregation period and are indicated as
DA1, DA3, DA6, DA9, and DA12 (Figure 2).

### 3.1.3 Data de-trending

Data stationarity is typically assumed when modelling. However, the present study uses crop
yield, which is non-stationary in nature. The crop yield depends on factors that affect its trend,
such as drought, flood, cultivars, and its own management. Therefore, it is advisable to remove
short-term fluctuations in crop yield before constructing the model (Montesino Pouzols and
Lendasse, 2010).
Among the methods available to de-trend data, the 'first difference' method is popular due to
its simplicity. In this method, the trend is removed from the time series by subtracting the
previous value $x^*(t-1)$ from the current one $x^*(t)$, as shown in Eq. 3. The de-trended value for
the first time step ($t = 1$) is not calculated. The length of the de-trended time series is $n = m-1$,
where $m$ is the length of the original time series. The de-trended data $x(t)$ has the same units as
the original data $x^*(t)$.
$$x(t) = x^*(t) - x^*(t-1) \qquad \text{(Eq. 3)}$$



Once the trend is removed, all the steps for constructing the ML models are carried out with
the de-trended time series. After the ML models are built, the de-trending procedure must be
applied in reverse after calculating a new prediction $x(t+1)$ to have that prediction in the
magnitude to the original time series. The reverse de-trending procedure can be done with Eq.
4, which is the solution for Eq. 3 for the de-trended prediction $x(t+1)$. In practical terms, the
prediction $x^*(t+1)$ in the original magnitude is calculated by adding the de-trended prediction
$x(t+1)$ to the last value of the original time series, i.e. $x^*(t)$.
$$x^*(t+1) = x^*(t) + x(t+1) \qquad\qquad (\text{Eq. 4})$$
The trend of the CY and DA time series was removed with Eq. 3. As can be observed, the
method for removing the trend does not generate the value for the first time step; therefore, the
de-trended CY data corresponds to the period 1967-2015 (49 years).
In the case of DA, Eq. 3 was applied as follows. Because the DA data is monthly, i.e. 12 values
per year, and CY data is seasonal, i.e. one value per year, the DA time series were extracted
and organised for each month from January to December to match them with the CY data
(Figure 2). This extraction/organisation procedure was carried out for each of the five
aggregation periods DA1, 3, 6, 9 and 12 months. A total of 60 DA time series ($12 \times 5$) were
obtained. To refer to these time series, a number (suffix) was added to indicate the month. In
this way, for example, the time series DA3_7 indicates the drought areas for July calculated
from the drought indicator with 3-month aggregation period. Eq. 3 for the removal of the trend
was applied to each of the 60 DA time series (Figure 2). The DA time series run from 1901 to
2015. For the construction of the ML models, the common period 1967-2015 (49 years) was
chosen.
**3.2 Step 2. Input variable selection**
In an ML model, the input, known as the predictor, is generally made up of independent
variables. These input variables are often arranged or aggregated in different ways to determine
the best model input representation. An example of arrangement is by considering different
previous time steps of the input variable, such as $t-1$ (the previous time), $t-2$, and so on.
Another way is by aggregating the input variable in different periods. For instance, when using
drought indicators as the predictors (input), the aggregation periods include 3, 6, 9, 12, and 24
months. Other aggregations include the average, or other statistics, over a period. In this step,
the idea is not to include all the variables and all their different possible arrangements or
aggregations but rather to choose the suitable input variables and discard those that do not
contribute significantly to the model's results.





There are different methods for selecting input variables. Based on the procedure, these
methods are classified into model-based and filter types (May et al., 2011). The model-based
type includes all those where the model runs and based on its performance, a specific variable
is chosen or discarded. The filter type includes methods where the variable is chosen *a priori*
through a generally statistical process and does not require the model to be run. Correlation
analysis, which falls under the second category, is often chosen for its simplicity and wide
application. Correlation is calculated between the time series of the output variable (CY in this
case) and the different input variables, including their various arrangements or aggregations.
In this study, for the selection of the relevant input variables, the correlation analysis was done.
The correlation was calculated between the de-trended time series of the seasonal CY and the
60 DAs (Figure 2). As mentioned before, due to DAs are monthly and CY is seasonal, 12 time
series of DAs were prepared, one per month, for each aggregation period. The DAs were then
correlated with the CY. Another option could be to use the yearly average value of the DAs,
such as the average of the DAs of the months of the cultivation period, or something similar.
However, we opted to identify the DAs of the months that have the highest correlation with the
seasonal CY and use them as inputs.
The approach of the selection of the most correlated DAs was chosen for two main reasons.
First, on the one hand, rice responds to the climate variations differently from one growth stage
to another over the year, which could be better captured with the information of some months
than others. On the other hand, different types of drought (i.e. meteorological, agricultural, and
hydrological) are expected to affect (impact) the crop yield to varying degrees throughout the
different stages of crop growth. This level of impact could be taken into account either by using
different hydro-meteorological variables or selecting different aggregation periods of the
meteorological variables, as in this case. An average of DAs could "hide" a significant drought
area that could contribute more (or less) to the final crop yield.
Second, in this research, ML models were built to be used at different stages of crop cultivation,
i.e. models to be applied in June, July, and so on, each of them with a different expected degree
of accuracy. Therefore, the use of time series for each month extracted from the DAs for all
the different aggregation periods (1, 3, 6, 9, and 12 months) is more appropriate than the
average (Figure 2).
Based on the correlation coefficient, the input variables were chosen. In total, 15 sets of input
variables (Table 2) were selected. Each set is made up of the different DA time series, i.e. DA1,
3, 6, 9, and 12. The number of variables is different in each set. These sets of input variables
are presented in the results section. All sets also include the de-trended CY from the previous



year (CY$_{t-1}$). CY$_{t-1}$ was used because, in the particular case of the study area, CY of the current
year is planned to be reached based on data of the previous year. The ML models were built
for each month (from January to December). The sets of inputs presented in Table 2 (Sect. 4.2)
indicate which time series of DAs have to be considered for the ML model's construction. The
models were built for each of the 15 input sets, more details are in the following sections. It
should be noted that for each month the DAs are those corresponding to the same month.


**Figure 2** Diagram showing how time series of monthly drought areas (DAs) are extracted and organised to match
them with the seasonal crop yield (CY) data. For each year there are 12 DA values and one CY value. DAs were
calculated for the aggregation periods 1, 3, 6, 9, and 12 months (DA1 to DA12). DAs were extracted and organised
by month, from January to December. For each month, the procedures of data de-trending, correlation, input
variable selection, and ML models construction were carried out. The entire flow was conducted for each of the
three regions analysed.



**3.3 Step 3. Polynomial regression models calculation**
For the case of PR, four types of models were tested (Table 1). All the PR models were built
for each month from January to December following Eq. 5 to 8. A total of 15 sets of
combinations of input variables were tested in each PR model. The best PR model was
identified for each month following the RMSE criterion (Eq. 9). MATLAB software was used
for implementation.
PR is an extension of linear regression that allows the use of more than one input variable to
calculate the output variable (Eq. 4).
$$y = b_0 + \sum_{i=1}^{n} b_i x_i + e \qquad \text{(Eq. 4)}$$
In Eq. 4, $y$ is the output variable, also known as the response, which in this case is the crop
yield. The term $x_i$ is the $i$-th input variable (predictor) from a total of $n$ variables. The regression
coefficients vector is represented by $b$. From the coefficients vector, $b_0$ is known as the
intercept. The vector of errors is indicated by $e$.
Table 1 shows four formulations of PR. The PR models are indicated as linear, pure-quadratic,
quadratic, and interactions. Descriptions of each and their equations are presented in Table 1
(Eq. 5 to 8). The input variable ($x_i$) was selected based on the correlation analysis (Sect. 2.2).
**Table 1** Polynomial regression (PR) types followed in this study.

| PR type | Equation | Description |
|---|---|---|
| Linear | (Eq. 5) $y = b_0 + \sum_{i=1}^{n} b_i x_i$ | It has an intercept and linear terms of predictors |
| Pure-quadratic | (Eq. 6) $y = b_0 + \sum_{i=1}^{n} b_i x_i + \sum_{i=1}^{n} b_{n+i} x_i^2$ | It has an intercept, as well as linear and squared terms of predictors |
| Quadratic | (Eq. 7) $y = b_0 + \sum_{i=1}^{n} b_i x_i + \sum_{i=1}^{n} b_{n+i} x_i^2 + \sum_{i=1}^{n-1} \sum_{j=i+1}^{n} b_{2n+(i-1)n-\frac{(i-1)i}{2}+(j-i)} x_i x_j$ | It has an intercept, linear and squared terms and all products of pairs of distinct predictors |
| Interactions | (Eq. 8) $y = b_0 + \sum_{i=1}^{n} b_i x_i + \sum_{i=1}^{n-1} \sum_{j=i+1}^{n} b_{n+(i-1)n-\frac{(i-1)i}{2}+(j-i)} x_i x_j$ | It has an intercept, linear terms of predictors, all products of pairs of distinct predictors and no squared terms |





The best PR model was identified from four types using the root mean square error (RMSE)
criterion. The RMSE is calculated between the observations (*o*) and the predictions (*p*), as
shown in Eq. 9. RMSE is one of the most widely used criteria in the comparison of observations
and model calculations.
$$\text{RMSE} = \sqrt{\frac{\sum_{i=1}^{n} (o_i - p_i)^2}{n}}$$   (Eq. 9)
**3.4 Step 4. Artificial neural network models calculation**
ANN is a method loosely based on imitating the basic functionality of neurons (i.e. the working
units of the human brain) (Govindaraju, 2000; Maier and Dandy, 2000). The input variables
(predictors) are connected to each other through mathematical formulations that allow complex
non-linear relationships to be represented. These connexions are symbolised as nodes
interconnected within a network aimed at calculating the output variable (response).
Of the different proposed ANN architectures (network designs), one of the most widely used
is the feedforward neural network (FFNN). The FFNN is schematised by a series of nodes
located in one of three layers: input, hidden or output. The number of input nodes is equal to
the number of input variables in the input layer (Elshorbagy et al., 2010). This first layer is in
turn connected to the hidden layer, which receives this name because the connections made
there may not be immediately evident to the model performer. In this hidden layer, the number
of nodes is not defined by default; rather, the greater the number of nodes, the more complex
the model. Finally, the nodes of the hidden layer are connected to those of the output layer. In
a single-output variable problem, there is only one node. ANNs are typically trained by non-
linear optimisation gradient-based algorithms, e.g. the Levenberg-Marquardt algorithm.
In the ANN setup, the number of nodes of the input layer was equal to the number of variables
of the respective combination. The number of nodes in the output layer was one and
corresponded to the seasonal crop production (CY). An iteration optimisation procedure was
carried out regarding the hidden layer, varying the number of nodes from 1 to 10. For each
number of nodes, 100 iterations were done, being 1,000 in total. For reproducibility of the
results, the random values were set to default at the beginning of the number of nodes change.
For each month, from January to December, the ANNs were built. MATLAB software was
used to implement the ANNs with the Levenberg-Marquardt algorithm for training. In each of
the ANNs, 85 % of the data was used for training-validation, and the rest for testing
(verification). The best model corresponding to each number of hidden nodes was identified,
i.e. ten models per month and the best model for each month. RMSE was used to identify the



best models. RMSE was calculated for (1) the training-validation dataset (RMSE_cal), (2) the
testing dataset (RMSE_test), and (3) the entire period (RMSE). In all the cases, the final (best)
model was chosen based on RMSE for the entire period. The iteration optimisation procedure,
including the calculation of RMSE, was carried out for each of the 15 sets of input variables
(Table 2) and for each month (Sect. 4.2).
**3.5 Step 5. Models application and combination**
Once the best ML models, PR and ANN, were known, the pair of models were selected for
each month. Depending on the performance of these models (and experience of their use), they
can be used either separately or combined, e.g. being run in parallel so that a modeller could
see the cases when models produce different results. An alternative is to use a dynamic
weighting of the models' outputs (e.g. with the weights being proportional to the historical
performance) to form a "model committee".
**4 Results and discussion**
**4.1 Data preparation: drought areas and crop yield**
Figure 3 shows the drought areas calculated for the three regions. In this heat map, columns
indicate the months and rows point out the years. The redder the colour, the larger the drought
area. In general, region 1 (Figure 3, the upper panel) presents the highest values concerning the
other two regions. In general, the 1990s show higher values of areas with respect to the rest of
the period, which agrees with Guha-Sapir (2019); in this decade, there were three droughts,
1993, 1996 and 2000. At the beginning of the period, large areas are also observed in the threee
regions; these results align with Bhalme and Mooley (1980).
In Figure 3, a pattern is observed in the drought areas distribution for all the aggregation
periods, i.e. from DA1 to DA12. In DA1, the areas mainly concentrate in the first months; even
the December column is almost white (without drought). Later, for DA3, the large areas are
located from April to November. Successively, for DA6 and DA9, the largest areas are
concentrated in the second half of the year. There are even droughts that end in the following
year; they are the reddish lines that are observed in the first semester (first columns). Finally,
in DA12, there are consecutive large areas indicated by the reddish lines; droughts usually
begin in the second semester and extend until the following year. These results show the
importance of considering more than one period of aggregation when using indicators based
on meteorological variables; each aggregation period can be a proxy for analysing different
types of drought and its effects.



**Figure 3** Drought areas (DAs) for each aggregation period (1, 3, 6, 9, and 12 months) and region. Top, middle, and bottom panels indicate region 1 (Bihar and Jharkhand), region 2 (West Bengal) and region 3 (Odisha).

Figure 4 shows the time series of de-trended CY and DA for the three regions. In the case of
DA (indicated in red), the values are displayed in inverse order to facilitate interpretation. In
general, when drought areas increase, this is expected to affect crop yield (decreasing).
Otherwise, when the drought area decreases, this effect favours an increase in crop yield. In
general, for the three regions, the decreases in CY coincide with the increases in DA. The
general pattern regarding DA variations is as follows. The values fluctuate throughout the year
for the aggregation periods of one and three months (DA1 and DA3). Subsequently, for DA6
to DA12, the values are concentrated in the second half of the year. These results also show
the usefulness of the different aggregation periods to capture different types of drought. The
effect of increasing DA seems not to be observed in decreasing CY for all cases of DAs. For
example, in region 1 (Figure 4, the upper panel), the decrease in 2004, one of the maximums,
does not coincide with increases in DA9 and DA12, but it does for DA1, DA3 and DA6. These
results also support the use of the different aggregation periods on drought assessments.
**4.2 Input variable selection (correlation analysis)**
Figure 5 summarises the correlation between the de-trended CY and the DAs, and Figure 6
presents the correlation for each monthly DA time series.
Figures 5 and 6 show that the correlation is different over the year in the three regions. In all
cases, the correlation coefficient increases until a maximum and then decreases. The month in
which the maximum value is reached is different for each region but falls within the crop season
(i.e. June to November/December). For region 1, it is in July. For region 2, there are four
months with this pattern, June, July, October, and November. Finally, for region 3, it is
October, November, and December.
These results of correlation can be useful for monitoring agricultural drought. For example, in
region 1, the drought areas calculated from SPEI6 (i.e. DA6) show a maximum correlation in
July. This correlation value means that the previous six months' accumulated effect is crucial
for the crop yield of the Kharif season, which covers more or less from June to
November/December.
Figure 5 shows the following pattern. In general, for region 1, results similar to DA6 are
observed for DA3, 9, and 12. For region 2, a similar pattern happens in the peaks, but in this
case two, one corresponding to DA1 and 3, and the other to DA6, 9, and 12. The first peak of
DA1 and DA3 may indicate that it is crucial to pay attention to the immediate period conditions
of one to three months. In the case of the second peak, the medium and long-term conditions,
6 to 12 months, are more important to monitor for the harvest month. For region 3, the peak





occurs at the end of the growing season, in almost all cases. Hence, the condition before the
growing season is decisive for the crop yield.

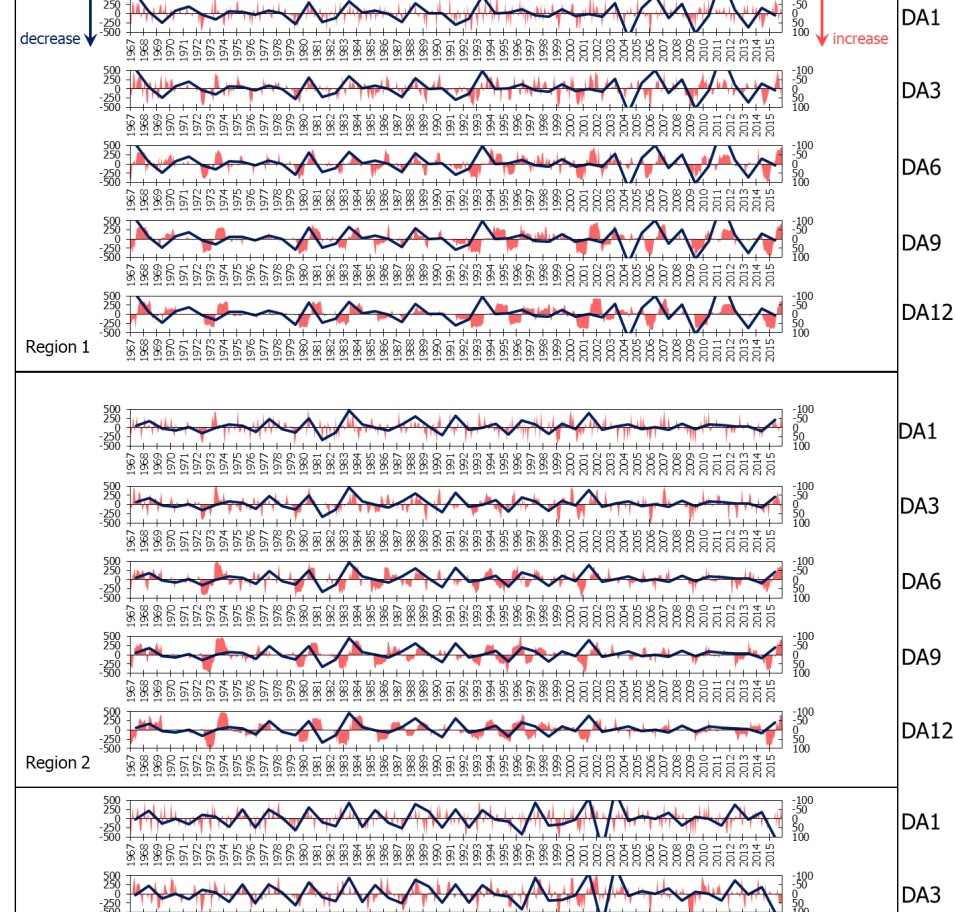

**Figure 4** Time series of the de-trended crop yield (CY) and drought areas (DAs) for each aggregation period (1,
3, 6, 9 and 12 months) and region. Top, middle, and bottom panels indicate region 1 (Bihar and Jharkhand), region
2 (West Bengal) and region 3 (Odisha).
Figure 6 shows how the correlation coefficients between CY and DA are positive outside the
growing season and negative within that season. However, this pattern is less evident for DA1
and DA3. The pattern shown by the correlation coefficients in Figure 6 supports the idea that
drought is an important factor in crop yield since the months with less drought are more
correlated with the increase in CY, and the months with more drought do so with decrease in
CY.
Figure 5 (d) shows the percentage of irrigated and rain-fed agriculture. For regions 1 and 2,
about half is by irrigation, while in region 3, only 35%. Perhaps this percentage of irrigation
for region 3 explains why the correlation coefficients for this region are higher than for the
other two (Figure 5, and 6 (c)). Region 3 is more dependent on rain for agriculture; therefore,
this condition is best captured when calculating drought with the precipitation, as in this case
(Sect. 3.2).

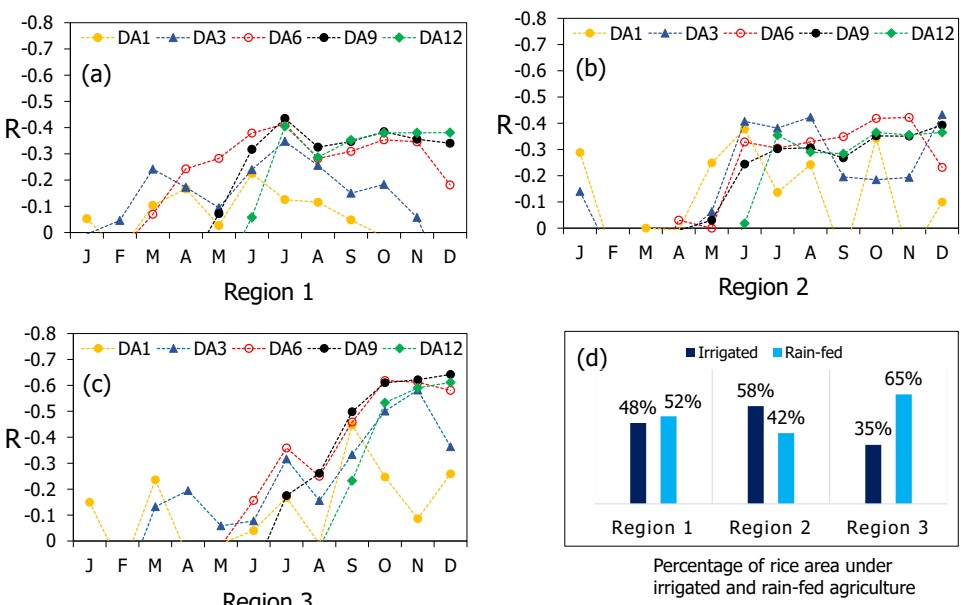


**Figure 5** Summary of correlation between de-trended crop yield (CY) and drought areas (DAs) for each
aggregation period (1, 3, 6, 9, and 12 months) and region: (a) region 1 (Bihar and Jharkhand), (b) region 2 (West
Bengal) and (c) region 3 (Odisha). Negative R indicates the correlation between the increase in DA and the
decrease in CY. Percentage of rice area under irrigated and rein-fed agriculture (d). Source of irrigated and rein-
fed agriculture data: Directorate of Rice Development (DRD), (2014).





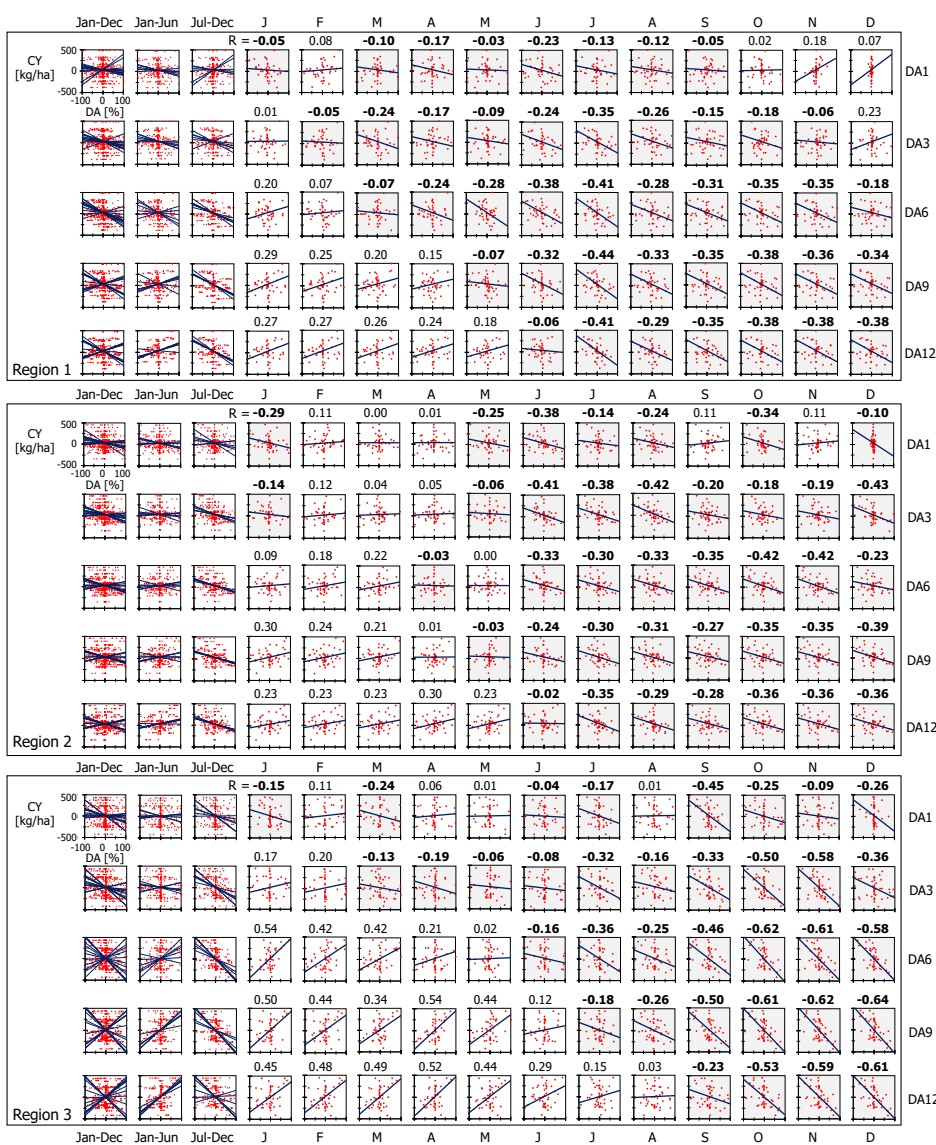

**Figure 6** Correlation (R) between de-trended crop yield (CY) and drought areas (DAs) for each aggregation period (1, 3, 6, 9, and 12 months) and region. DA is on the *x*-axis, and CY is on the *y*-axis. Results are shown for each monthly DA time series from June to December (J to D). Top, middle, and bottom panels indicate region 1 (Bihar and Jharkhand), region 2 (West Bengal), and region 3 (Odisha). Negative R indicates the correlation between increase in DA and decrease in CY.





Figure 5 (a, b, and c) shows the following pattern in the three regions. The correlation
coefficients between CY and DAs increase according to the aggregation periods and the month
of analysis. DA1 and DA3 have a better correlation in the first months of the year. DA6 has a
better correlation in the subsequent months, between May and June. Finally, DA9 and 12 do
so within the second half of the year.
Each respective DA time series reaches a maximum (or maximums) of correlation, and then
correlation decreases. According to this pattern, the 15 combinations of input variables shown
in Table 2 were selected. As earlier mentioned, the CY of the previous year was included in all
combinations and is indicated as $CY_{t-1}$. Combinations 1 to 5 only include a DA time series.
Combinations 6 to 9 are DA pairs that were calculated with the drought indicator of successive
aggregation times. For example, combination 6 forms DA1 and 3, combination 7 includes DA3
and 6, and so on. Similarly, combinations 10 to 13 are proposed, but for triples. Combinations
13 and 14 are fourfold. Finally, the last combination (15th) is made up of all the DA series.
As mentioned, the models were built for each month (January to December) using the 15
combinations (Table 2) in each case. For example, for the case of January the monthly series
of DAs extracted for January were used. These DAs are DA1_1, DA3_1, DA6_1, DA9_1, and
DA12_1. The suffix indicates the month. Then, the different DA1_1 to DA12_1 were used
following the 15 combinations shown in Table 2 to build the ML models (ANN and PR) for
January. Similarly, it was carried out from February to December.
**Table 2** Input sets (combinations) to build the ML models. CY and DA stand for crop yield and drought area.
DAs are calculated with the drought indicator for the aggregate period of 1, 3, 6, 9, and 12 months (details in Sect.

4.2).

| Input set (combination) | Input variables |
| --- | --- |
| 1 | $CY_{t-1}$, DA1 |
| 2 | $CY_{t-1}$, DA3 |
| 3 | $CY_{t-1}$, DA6 |
| 4 | $CY_{t-1}$, DA9 |
| 5 | $CY_{t-1}$, DA12 |
| 6 | $CY_{t-1}$, DA1,3 |
| 7 | $CY_{t-1}$, DA3,6 |
| 8 | $CY_{t-1}$, DA6,9 |
| 9 | $CY_{t-1}$, DA9,12 |
| 10 | $CY_{t-1}$, DA1,3,6 |
| 11 | $CY_{t-1}$, DA3,6,9 |
| 12 | $CY_{t-1}$, DA6,9,12 |
| 13 | $CY_{t-1}$, DA1,3,6,9 |
| 14 | $CY_{t-1}$, DA3,6,9,12 |
| 15 | $CY_{t-1}$, DA1,3,6,9,12 |



### 4.3 ANN and PR models

The results show different magnitudes of error between the observed and predicted CY. The models with the lowest error are presented in Figures 7, 8 and 9, for each of the three regions. The pair of ANN and PR that best predicts CY is shown for each month. The RMSE is also indicated in each case. On the other hand, Figure 10 shows the error for each input set (combination); the lowest error achieved in each month is presented in each case both for each ANN and PR.

In general, ANN shows the least errors, as expected (Figure 10). However, the results of PR are not much worse compared to those of ANN; for example, in some cases, the errors shown by linear PR are very close to those of ANN (e.g. Figure 10, region 2). In general, it is observed that the models with the lowest errors correspond to region 2, followed by region 3 and region 1 (Figure 10). It is attributed to the different degrees of crop irrigation with surface and mostly groundwater, which determines the accuracy of the modelling in the different regions. Another factor contributing to the models' performance is the drastic changes in the CY data, where regions 1 and 3 are the ones that presented the most, and to a much lesser extent, region 2.

Figure 10 shows that in the three regions, different types of PR showed better results. In general, linear and pure-quadratic indicate more stable results (no sudden changes among the different realisations) but not better than quadratic and interactions. In general quadratic and interactions present better results, being in some cases very close to those shown by ANN, e.g. PR interactions (Figure 10, region 1).

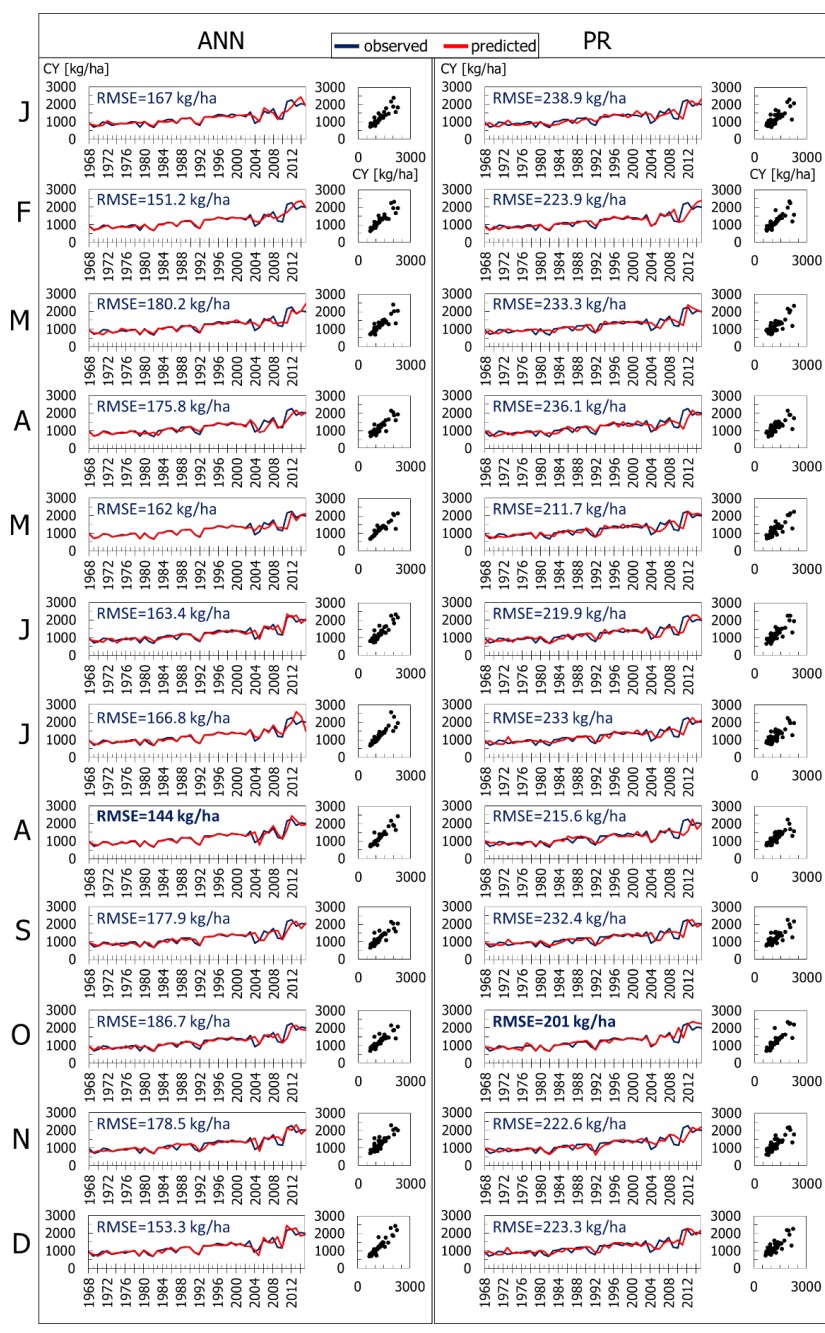

**Figure 7** ANN and PR models for predicting seasonal crop yield (CY) built for each time series of monthly drought areas (DAs): region 1 (Bihar and Jharkhand). The model with the lowest error (RMSE) is presented for each month, from January to December (J to D).



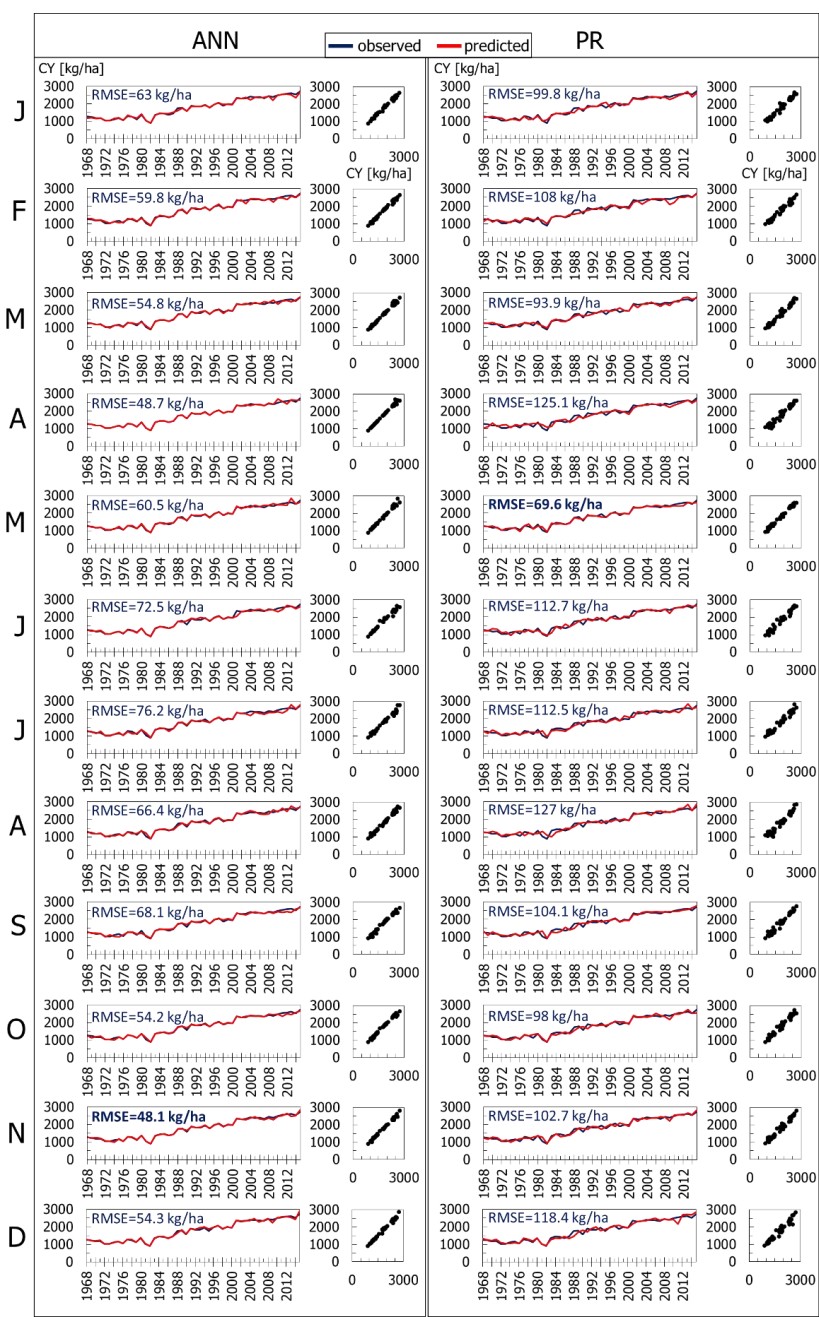

**Figure 8** ANN and PR models for predicting seasonal crop yield (CY) built for each time series of monthly drought areas (DAs): region 2 (West Bengal). The model with the lowest error (RMSE) is presented for each month, from January to December (J to D).

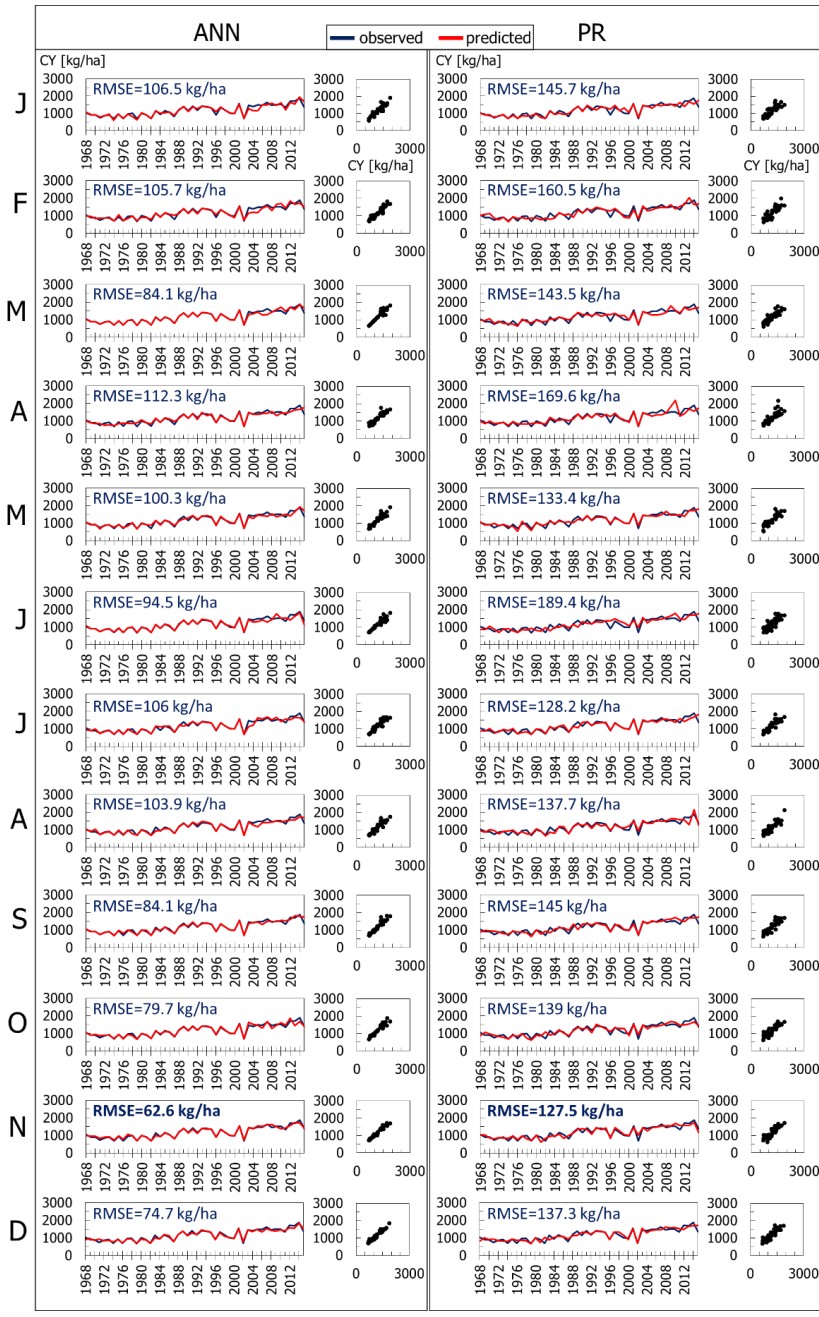

**Figure 9** ANN and PR models for predicting seasonal crop yield (CY) built for each time series of monthly drought areas (DAs): region 3 (Odisha). The model with the lowest error (RMSE) is presented for each month, from January to December (J to D).




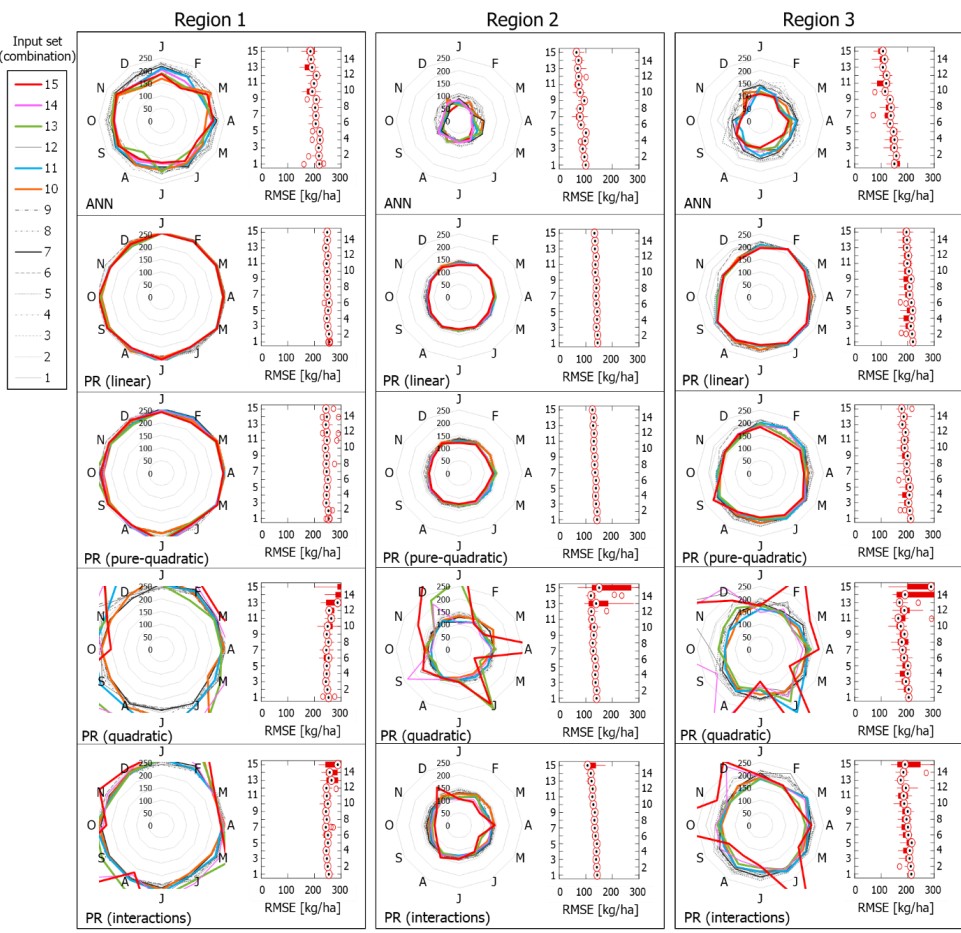

**Figure 10** Root mean square error (RMSE) [kg/ha] for each of the 15 input sets (combinations) of the ANN and PR models built for each region. For each set of input (from one to 15), the lowest errors are presented for each month (January to December). Results of each input set are shown with lines to facilitate the analysis. Left, middle, and right panels indicate region 1 (Bihar and Jharkhand), region 2 (West Bengal) and region 3 (Odisha).

### 4.4 Models application and combination

The best performing models were selected for each month. Table 3 shows the summary of these models, which includes the input set (combination), number of nodes, and errors for ANN, and input set, type, and errors for PR. The number of nodes indicates the degree of non-linearity presented in each model. In this way, the more nodes, the more complex the model is in the case of ANN. On the other hand, quadratic and interactions are the types that showed the best performance in PR models. In all cases, within the combinations of input variables, a single DA time series corresponding to one of the various aggregation periods (D1, D3, D6, D9 or D12) that by itself produced good results was not found. The input sets are made up of two and





up to six different DAs corresponding to the various aggregation periods. Thus, using more than
one aggregation period of drought indicator results in better model performance.
Tables 4, 5 and 6 are derived from Table 3. These three tables show the PR formulas for region 1,
2 and 3, respectively. In each table, the PR formula and the inputs are indicated. These formulas
are also intended to be a stand-alone tool in the CY prediction for each region.
The application of PR models begins by selecting the formula of the PR model (Table 4, 5, or 6).
For example, in the case of region 1, if the drought indicator data is available up to March (including
it), the formula for March is chosen from Table 4. After, DAs are calculated (Sect. 3.1.2), and the
time series of DAs are updated. According to Table 4, the DA1 and DA3 are required in this
example. Then, from these time series of DAs, values of March are extracted, i.e. DA1_3 and
DA3_3 (see Sect. 3.2 and 4.2). Then, the de-trending procedure is applied to each time series (Sect.
3.1.3). After, the CY is calculated. Finally, the reverse de-trending procedure is carried out to have
the predicted CY in the same order of magnitude as the original CY data (Sect. 3.1.3). At the same
time, or when it can be computed, the ANN model of the month under analysis is applied.
**Table 3** Summary of the ANN and PR models for predicting crop yield (CY) built for each month and region: (1)
Bihar and Jharkhand, (2) West Bengal, and (3) Odisha. The table shows the models built with the lowest error
(RMSE). DA stands for drought area.

| | | ANN | | | | PR | | | |
|---|---|---|---|---|---|---|---|---|---|
| Region | Month | Input set (combination) | No. nodes | RMSE [kg/ha] | Month | Input set (combination) | Type | RMSE [kg/ha] | |
| | Jan | 10 | $CY_{t-1}$, DA1,3,6 | 4 | 167.0 | Jan | 8 | $CY_{t-1}$, DA6,9 | quadratic | 238.9 |
| | Feb | 15 | $CY_{t-1}$, DA1,3,6,9,12 | 6 | 151.2 | Feb | 13 | $CY_{t-1}$, DA1,3,6,9 | quadratic | 223.9 |
| | Mar | 11 | $CY_{t-1}$, DA3,6,9 | 7 | 180.2 | Mar | 6 | $CY_{t-1}$, DA1,3 | quadratic | 233.3 |
| | Apr | 10 | $CY_{t-1}$, DA1,3,6 | 9 | 175.8 | Apr | 15 | $CY_{t-1}$, DA1,3,6,9,12 | interactions | 236.1 |
| | May | 15 | $CY_{t-1}$, DA1,3,6,9,12 | 5 | 162.0 | May | 10 | $CY_{t-1}$, DA1,3,6 | quadratic | 211.7 |
| Region 1 | Jun | 13 | $CY_{t-1}$, DA1,3,6,9 | 2 | 163.4 | Jun | 10 | $CY_{t-1}$, DA1,3,6 | interactions | 219.9 |
| | Jul | 15 | $CY_{t-1}$, DA1,3,6,9,12 | 10 | 166.8 | Jul | 6 | $CY_{t-1}$, DA1,3 | quadratic | 233.0 |
| | Aug | 13 | $CY_{t-1}$, DA1,3,6,9 | 5 | 144.0 | Aug | 15 | $CY_{t-1}$, DA1,3,6,9,12 | interactions | 215.6 |
| | Sep | 6 | $CY_{t-1}$, DA1,3 | 5 | 177.9 | Sep | 7 | $CY_{t-1}$, DA3,6 | quadratic | 232.4 |
| | Oct | 14 | $CY_{t-1}$, DA3,6,9,12 | 6 | 186.7 | Oct | 15 | $CY_{t-1}$, DA1,3,6,9,12 | quadratic | 201.0 |
| | Nov | 8 | $CY_{t-1}$, DA6,9 | 4 | 178.5 | Nov | 10 | $CY_{t-1}$, DA1,3,6 | interactions | 222.6 |
| | Dec | 10 | $CY_{t-1}$, DA1,3,6 | 4 | 153.3 | Dec | 13 | $CY_{t-1}$, DA1,3,6,9 | pure-quadratic | 223.3 |
| | Jan | 13 | $CY_{t-1}$, DA1,3,6,9 | 8 | 63.0 | Jan | 14 | $CY_{t-1}$, DA3,6,9,12 | quadratic | 99.8 |
| | Feb | 11 | $CY_{t-1}$, DA3,6,9 | 10 | 59.8 | Feb | 15 | $CY_{t-1}$, DA1,3,6,9,12 | interactions | 108.0 |
| | Mar | 7 | $CY_{t-1}$, DA3,6 | 8 | 54.8 | Mar | 15 | $CY_{t-1}$, DA1,3,6,9,12 | interactions | 93.9 |
| | Apr | 14 | $CY_{t-1}$, DA3,6,9,12 | 7 | 48.7 | Apr | 14 | $CY_{t-1}$, DA1,3,6,9,12 | interactions | 125.1 |
| | May | 15 | $CY_{t-1}$, DA1,3,6,9,12 | 10 | 60.5 | May | 15 | $CY_{t-1}$, DA1,3,6,9,12 | quadratic | 69.6 |
| Region 2 | Jun | 13 | $CY_{t-1}$, DA1,3,6,9 | 7 | 72.5 | Jun | 10 | $CY_{t-1}$, DA1,3,6 | quadratic | 112.7 |
| | Jul | 6 | $CY_{t-1}$, DA1,3 | 6 | 76.2 | Jul | 10 | $CY_{t-1}$, DA1,3,6 | quadratic | 112.5 |
| | Aug | 6 | $CY_{t-1}$, DA1,3 | 9 | 66.4 | Aug | 13 | $CY_{t-1}$, DA1,3,6,9 | interactions | 127.0 |
| | Sep | 6 | $CY_{t-1}$, DA1,3 | 10 | 68.1 | Sep | 15 | $CY_{t-1}$, DA1,3,6,9,12 | interactions | 104.1 |
| | Oct | 7 | $CY_{t-1}$, DA3,6 | 10 | 54.2 | Oct | 15 | $CY_{t-1}$, DA1,3,6,9,12 | interactions | 98.0 |
| | Nov | 7 | $CY_{t-1}$, DA3,6 | 10 | 48.1 | Nov | 15 | $CY_{t-1}$, DA1,3,6,9,12 | interactions | 102.7 |
| | Dec | 15 | $CY_{t-1}$, DA1,3,6,9,12 | 8 | 54.3 | Dec | 14 | $CY_{t-1}$, DA3,6,9,12 | interactions | 118.4 |
| | Jan | 15 | $CY_{t-1}$, DA1,3,6,9,12 | 7 | 106.5 | Jan | 14 | $CY_{t-1}$, DA3,6,9,12 | quadratic | 145.7 |
| | Feb | 13 | $CY_{t-1}$, DA1,3,6,9 | 10 | 105.7 | Feb | 10 | $CY_{t-1}$, DA1,3,6 | quadratic | 160.5 |
| | Mar | 15 | $CY_{t-1}$, DA1,3,6,9,12 | 9 | 84.1 | Mar | 12 | $CY_{t-1}$, DA6,9,12 | quadratic | 143.5 |
| | Apr | 15 | $CY_{t-1}$, DA1,3,6,9,12 | 4 | 112.3 | Apr | 14 | $CY_{t-1}$, DA3,6,9,12 | quadratic | 169.6 |
| | May | 12 | $CY_{t-1}$, DA6,9,12 | 10 | 100.3 | May | 15 | $CY_{t-1}$, DA1,3,6,9,12 | quadratic | 133.4 |
| Region 3 | Jun | 15 | $CY_{t-1}$, DA1,3,6,9,12 | 9 | 94.5 | Jun | 12 | $CY_{t-1}$, DA6,9,12 | quadratic | 189.4 |
| | Jul | 15 | $CY_{t-1}$, DA1,3,6,9,12 | 7 | 106.0 | Jul | 15 | $CY_{t-1}$, DA1,3,6,9,12 | quadratic | 128.2 |
| | Aug | 12 | $CY_{t-1}$, DA6,9,12 | 7 | 103.9 | Aug | 15 | $CY_{t-1}$, DA1,3,6,9,12 | interactions | 137.7 |
| | Sep | 11 | $CY_{t-1}$, DA3,6,9 | 9 | 84.1 | Sep | 13 | $CY_{t-1}$, DA1,3,6,9 | quadratic | 145.0 |
| | Oct | 15 | $CY_{t-1}$, DA1,3,6,9,12 | 10 | 79.7 | Oct | 10 | $CY_{t-1}$, DA1,3,6 | quadratic | 139.0 |
| | Nov | 11 | $CY_{t-1}$, DA3,6,9 | 10 | 62.6 | Nov | 10 | $CY_{t-1}$, DA1,3,6 | quadratic | 127.5 |
| | Dec | 11 | $CY_{t-1}$, DA3,6,9 | 9 | 74.7 | Dec | 8 | $CY_{t-1}$, DA6,9 | quadratic | 137.3 |






**Table 4** PR models for predicting crop yield (CY) built for each month: region 1 (Bihar and Jharkhand). For each
moth, it is indicated the input (x1 to x6) and the PR formula. DA stands for drought area.

| Month | Input | | | | | | PR model |
|---|---|---|---|---|---|---|---|
| | $x_1$ | $x_2$ | $x_3$ | $x_4$ | $x_5$ | $x_6$ | |
| Jan | $CY_{t-1}$ | DA6 | DA9 | | | | $-60.7111 -0.1944x_1 -0.2201x_2 +1.2033x_3 -0.0023x_1x_2 +0.0043x_1x_3 -0.0372x_2x_3$ $+0.0003x_1^2 +0.0504x_2^2 +0.0308x_3^2$ |
| Feb | $CY_{t-1}$ | DA1 | DA3 | DA6 | DA9 | | $-27.4716 -0.4688x_1 +1.8718x_2 -1.3313x_3 -0.2611x_4 +1.3878x_5 -0.0137x_1x_2$ $+0.0135x_1x_3 +0.0032x_1x_4 +0.0064x_1x_5 +0.0823x_2x_3 +0.0574x_2x_4 +0.0935x_2x_5$ $-0.0544x_3x_4 -0.0746x_3x_5 -0.0241x_4x_5 +0.0014x_1^2 -0.0496x_2^2 -0.0202x_3^2 -0.0016x_4^2$ $+0.0227x_5^2$ |
| Mar | $CY_{t-1}$ | DA1 | DA3 | | | | $28.1213 -0.5204x_1 -0.4908x_2 +0.0545x_3 +0.0051x_1x_2 -0.0093x_1x_3 +0.0033x_2x_3$ $+0.0003x_1^2 -0.0107x_2^2 +0.0086x_3^2$ |
| Apr | $CY_{t-1}$ | DA1 | DA3 | DA6 | DA9 | DA12 | $-24.3419 -0.4785x_1 -0.1965x_2 -0.1356x_3 +0.0848x_4 -0.4774x_5 +0.8029x_6 +0.0066x_1x_2$ $+0.0031x_1x_3 -0.0128x_1x_4 +0.0081x_1x_5 -0.0003x_1x_6 +0.0067x_2x_3 -0.0604x_2x_4$ $+0.1495x_2x_5 -0.0169x_2x_6 +0.0248x_3x_4 -0.1295x_3x_5 -0.0306x_3x_6 +0.0458x_4x_5$ $+0.0516x_4x_6 +0.0595x_5x_6$ |
| May | $CY_{t-1}$ | DA1 | DA3 | DA6 | | | $113.2521 -0.5132x_1 +1.0101x_2 -1.4019x_3 -1.1130x_4 +0.0100x_1x_2 +0.0150x_1x_3$ $-0.0027x_1x_4 +0.0250x_2x_3 -0.0655x_2x_4 +0.0596x_3x_4 -0.0006x_1^2 -0.0358x_2^2 -0.0380x_3^2$ $-0.0495x_4^2$ |
| Jun | $CY_{t-1}$ | DA1 | DA3 | DA6 | | | $54.3 -0.3715x_1 +1.4832x_2 +0.1432x_3 -3.0648x_4 -0.0106x_1x_2 +0.0256x_1x_3 -0.0111x_1x_4$ $-0.0556x_2x_3 +0.0648x_2x_4 -0.0172x_3x_4$ |
| Jul | $CY_{t-1}$ | DA1 | DA3 | | | | $18.7237 -0.3166x_1 +1.3310x_2 -3.0099x_3 -0.0030x_1x_2 +0.0024x_1x_3 +0.0054x_2x_3$ $+0.0001x_1^2 +0.0065x_2^2 -0.0065x_3^2$ |
| Aug | $CY_{t-1}$ | DA1 | DA3 | DA6 | DA9 | DA12 | $59.2373 -0.6972x_1 +0.1791x_2 +5.1900x_3 -1.3783x_4 -6.9753x_5 +1.5471x_6 -0.0142x_1x_2$ $+0.0072x_1x_3 +0.1163x_1x_4 -0.1285x_1x_5 +0.0294x_1x_6 -0.3670x_2x_3 +0.0897x_2x_4$ $+0.2332x_2x_5 +0.0922x_2x_6 +0.3014x_3x_4 +0.3444x_3x_5 -0.4160x_3x_6 -0.5819x_4x_5$ $-0.0450x_4x_6 +0.3299x_5x_6$ |
| Sep | $CY_{t-1}$ | DA3 | DA6 | | | | $44.8563 -0.4565x_1 +0.6884x_2 -1.9466x_3 +0.0053x_1x_2 -0.0005x_1x_3 +0.0012x_2x_3$ $+0.0004x_1^2 -0.0172x_2^2 -0.0002x_3^2$ |
| Oct | $CY_{t-1}$ | DA1 | DA3 | DA6 | DA9 | DA12 | $76.1546 +0.0046x_1 -2.2220x_2 +1.0816x_3 +19.1690x_4 -53.2338x_5 +29.1398x_6$ $+0.0048x_1x_2 +0.0155x_1x_3 -0.0383x_1x_4 -0.0868x_1x_5 +0.1254x_1x_6 -0.0444x_2x_3$ $+0.0448x_2x_4 +0.0175x_2x_5 -0.0552x_2x_6 +0.2154x_3x_4 -1.0260x_3x_5 +0.7776x_3x_6$ $+3.2060x_4x_5 -3.3267x_4x_6 +11.6655x_5x_6 +0.0002x_1^2 -0.0547x_2^2 +0.1171x_3^2 +0.2874x_4^2$ $-7.7995x_5^2 -4.0845x_6^2$ |
| Nov | $CY_{t-1}$ | DA1 | DA3 | DA6 | DA9 | | $30.0286 -0.4536x_1 -0.6721x_2 -0.8270x_3 -7.0981x_4 +5.3007x_5 -0.0339x_1x_2$ $+0.0086x_1x_3 +0.0107x_1x_4 -0.0084x_1x_5 +0.1347x_2x_3 +0.1123x_2x_4 -0.0596x_2x_5$ $+0.2355x_3x_4 -0.2262x_3x_5 -0.0117x_4x_5$ |
| Dec | $CY_{t-1}$ | DA1 | DA3 | DA6 | DA9 | | $29.2005 -0.3816x_1 -0.6953x_2 +0.8469x_3 +1.2024x_4 -3.2563x_5 +0.0005x_1^2 -0.5339x_2^2$ $-0.0047x_3^2 -0.0119x_4^2 +0.0083x_5^2$ |












**Table 5** PR models for predicting crop yield (CY) built for each month: region 2 (West Bengal). For each moth,
it is indicated the input (x1 to x6) and the PR formula. DA stands for drought area.

| Month | Input | | | | | | PR model |
|---|---|---|---|---|---|---|---|
| | $x_1$ | $x_2$ | $x_3$ | $x_4$ | $x_5$ | $x_6$ | |
| Jan | $CY_{t-1}$ | DA3 | DA6 | DA9 | DA12 | | $8.5606 -0.2404x_1 -1.1236x_2 -0.7606x_3 +6.6535x_4 -5.3772x_5 +0.0087x_1x_2 -0.0044x_1x_3 -0.0182x_1x_4 +0.0234x_1x_5 +0.0080x_2x_3 +0.0234x_2x_4 -0.0037x_2x_5 -0.0402x_3x_4 +0.1648x_3x_5 +0.0200x_4x_5 +0.0001x_1^2 -0.0145x_2^2 -0.0657x_3^2 +0.0544x_4^2 -0.0952x_5^2$ |
| Feb | $CY_{t-1}$ | DA1 | DA3 | DA6 | DA9 | DA12 | $-24.8742 -0.5460x_1 -0.1190x_2 +0.2175x_3 +0.7776x_4 -8.6335x_5 +6.4022x_6 -0.0164x_1x_2 +0.0095x_1x_3 -0.0251x_1x_4 +0.0262x_1x_5 -0.0057x_1x_6 -0.0179x_2x_3 -0.0241x_2x_4 -0.1705x_2x_5 +0.1579x_2x_6 +0.0064x_3x_4 +0.2383x_3x_5 -0.2779x_3x_6 -0.0117x_4x_5 +0.0266x_4x_6 +0.0614x_5x_6$ |
| Mar | $CY_{t-1}$ | DA1 | DA3 | DA6 | DA9 | DA12 | $35.6904 -0.3835x_1 -0.9286x_2 +0.1960x_3 -0.3445x_4 -0.3559x_5 +0.6370x_6 -0.0025x_1x_2 -0.0009x_1x_3 +0.0111x_1x_4 -0.0252x_1x_5 +0.0144x_1x_6 -0.0059x_2x_3 +0.0426x_2x_4 +0.0063x_2x_5 +0.0012x_2x_6 -0.0362x_3x_4 -0.1287x_3x_5 -0.0038x_3x_6 +0.0242x_4x_5 -0.0355x_4x_6 +0.0394x_5x_6$ |
| Apr | $CY_{t-1}$ | DA3 | DA6 | DA9 | DA12 | | $8.5856 -0.1865x_1 +1.5824x_2 -1.0816x_3 -1.0256x_4 +1.7846x_5 -0.0164x_1x_2 +0.0242x_1x_3 -0.0013x_1x_4 +0.0009x_1x_5 -0.0084x_2x_3 +0.0073x_2x_4 -0.0710x_2x_5 -0.0430x_3x_4 +0.0659x_3x_5 +0.0317x_4x_5$ |
| May | $CY_{t-1}$ | DA1 | DA3 | DA6 | DA9 | DA12 | $-25.0101 -0.8233x_1 -1.8073x_2 +1.1145x_3 +1.6217x_4 +0.9651x_5 +0.5729x_6 +0.0254x_1x_2 -0.1198x_1x_3 +0.0959x_1x_4 -0.0112x_1x_5 +0.0311x_1x_6 -0.2178x_2x_3 +0.3465x_2x_4 -0.3214x_2x_5 +0.0602x_2x_6 -0.9192x_3x_4 +1.2301x_3x_5 -0.2167x_3x_6 -0.8955x_4x_5 +0.1015x_4x_6 +0.0662x_5x_6 +0.0048x_1^2 -0.0096x_2^2 +0.3527x_3^2 +0.4308x_4^2 -0.0492x_5^2 +0.0639x_6^2$ |
| Jun | $CY_{t-1}$ | DA1 | DA3 | DA6 | | | $90.7623 -0.5785x_1 +0.1582x_2 -2.7914x_3 +0.8655x_4 -0.0176x_1x_2 +0.0093x_1x_3 -0.0108x_1x_4 +0.0533x_2x_3 -0.0521x_2x_4 +0.1589x_3x_4 +0.0012x_1^2 +0.0072x_2^2 -0.0974x_3^2 -0.0714x_4^2$ |
| Jul | $CY_{t-1}$ | DA1 | DA3 | DA6 | | | $26.1164 -0.6892x_1 -0.6723x_2 -5.5280x_3 +4.6922x_4 +0.0070x_1x_2 +0.0111x_1x_3 -0.0148x_1x_4 -0.1301x_2x_3 +0.0838x_2x_4 +0.5157x_3x_4 +0.0014x_1^2 +0.0679x_2^2 -0.1671x_3^2 -0.3540x_4^2$ |
| Aug | $CY_{t-1}$ | DA1 | DA3 | DA6 | DA9 | | $55.6167 -0.2284x_1 -0.0182x_2 -1.7996x_3 -4.0674x_4 +3.7965x_5 +0.0117x_1x_2 -0.0259x_1x_3 +0.0556x_1x_4 -0.0484x_1x_5 -0.0176x_2x_3 -0.1459x_2x_4 +0.1017x_2x_5 -0.0487x_3x_4 +0.2346x_3x_5 -0.1273x_4x_5$ |
| Sep | $CY_{t-1}$ | DA1 | DA3 | DA6 | DA9 | DA12 | $35.6058 -0.3263x_1 +1.9755x_2 -0.4197x_3 -3.5963x_4 +2.7383x_5 -1.2234x_6 +0.0013x_1x_2 -0.0057x_1x_3 -0.0470x_1x_4 +0.0042x_1x_5 +0.0475x_1x_6 +0.0033x_2x_3 -0.1889x_2x_4 +0.0749x_2x_5 +0.1060x_2x_6 +0.0179x_3x_4 -0.0003x_3x_5 +0.0412x_3x_6 +0.0291x_4x_5 -0.0312x_4x_6 -0.0379x_5x_6$ |
| Oct | $CY_{t-1}$ | DA1 | DA3 | DA6 | DA9 | DA12 | $7.7675 -0.1875x_1 -0.1476x_2 -0.8333x_3 -5.1327x_4 +15.3857x_5 -10.6323x_6 -0.0012x_1x_2 -0.0011x_1x_3 +0.0588x_1x_4 +0.0365x_1x_5 -0.0886x_1x_6 -0.1339x_2x_3 +0.1763x_2x_4 -0.5955x_2x_5 +0.4854x_2x_6 -0.4231x_3x_4 -0.2159x_3x_5 +0.6868x_3x_6 +0.3521x_4x_5 +0.0666x_4x_6 -0.4145x_5x_6$ |
| Nov | $CY_{t-1}$ | DA1 | DA3 | DA6 | DA9 | DA12 | $38.3601 -0.2443x_1 +1.7236x_2 -0.6584x_3 -6.7484x_4 +13.3609x_5 -9.4895x_6 +0.0114x_1x_2 +0.0162x_1x_3 +0.0331x_1x_4 -0.0817x_1x_5 +0.0478x_1x_6 +0.0370x_2x_3 -0.1350x_2x_4 -0.0212x_2x_5 +0.1631x_2x_6 -0.1562x_3x_4 -0.0082x_3x_5 +0.1229x_3x_6 +0.2672x_4x_5 -0.0938x_4x_6 -0.1335x_5x_6$ |
| Dec | $CY_{t-1}$ | DA3 | DA6 | DA9 | DA12 | | $24.769 -0.1091x_1 -2.9747x_2 +2.9990x_3 -5.4144x_4 +3.3374x_5 +0.0083x_1x_2 -0.0069x_1x_3 +0.0596x_1x_4 -0.0630x_1x_5 +0.0755x_2x_3 +0.0127x_2x_4 +0.0094x_2x_5 -0.0052x_3x_4 -0.0884x_3x_5 +0.0361x_4x_5$ |












**Table 6** PR models for predicting crop yield (CY) built for each month: region 3 (Odisha). For each moth, it is
indicated the input (x1 to x6) and the PR formula. DA stands for drought area.

| Month | Input | | | | | | PR model |
|---|---|---|---|---|---|---|---|
| | $x_1$ | $x_2$ | $x_3$ | $x_4$ | $x_5$ | $x_6$ | |
| Jan | $CY_{t-1}$ | DA3 | DA6 | DA9 | DA12 | | $-149.3429 -0.4867x_1 -1.5749x_2 +2.0827x_3 +5.9761x_4 -6.0586x_5 -0.0022x_1x_2 +0.0100x_1x_3 +0.0200x_1x_4 +0.0045x_1x_5 -0.0142x_2x_3 -0.2414x_2x_4 +0.1392x_2x_5 -0.1332x_3x_4 +0.1123x_3x_5 +0.2083x_4x_5 +0.0022x_1^2 +0.0262x_2^2 +0.0771x_3^2 +0.0431x_4^2 -0.1405x_5^2$ |
| Feb | $CY_{t-1}$ | DA1 | DA3 | DA6 | | | $-90.6767 -0.6674x_1 +0.1283x_2 +0.2580x_3 +0.4540x_4 -0.0041x_1x_2 +0.0141x_1x_3 -0.0009x_1x_4 +0.0055x_2x_3 -0.0195x_2x_4 +0.0771x_3x_4 +0.0006x_1^2 +0.0313x_2^2 -0.0207x_3^2 +0.0129x_4^2$ |
| Mar | $CY_{t-1}$ | DA6 | DA9 | DA12 | | | $-168.6741 -0.7249x_1 +0.2079x_2 -2.2594x_3 +2.2421x_4 +0.0074x_1x_2 -0.0102x_1x_3 +0.0347x_1x_4 -0.0159x_2x_3 +0.0009x_2x_4 +0.1147x_3x_4 +0.0025x_1^2 +0.0454x_2^2 -0.0197x_3^2 +0.0318x_4^2$ |
| Apr | $CY_{t-1}$ | DA3 | DA6 | DA9 | DA12 | | $-116.7973 -0.6789x_1 -0.4066x_2 -0.5459x_3 +3.4428x_4 -3.2126x_5 +0.0008x_1x_2 -0.0110x_1x_3 +0.0063x_1x_4 +0.0337x_1x_5 +0.0647x_2x_3 -0.1280x_2x_4 +0.0847x_2x_5 -0.0041x_3x_4 -0.1576x_3x_5 -0.0357x_4x_5 +0.0025x_1^2 -0.0386x_2^2 +0.0180x_3^2 +0.0968x_4^2 +0.1431x_5^2$ |
| May | $CY_{t-1}$ | DA1 | DA3 | DA6 | DA9 | DA12 | $-56.0895 -0.8435x_1 -1.5688x_2 +5.5848x_3 -5.6556x_4 -0.0876x_5 -0.4449x_6 +0.0396x_1x_2 -0.0552x_1x_3 +0.0130x_1x_4 +0.0414x_1x_5 -0.0155x_1x_6 +0.0691x_2x_3 -0.1386x_2x_4 +0.4106x_2x_5 +0.0874x_2x_6 +0.2997x_3x_4 -0.2552x_3x_5 -0.4282x_3x_6 -0.0482x_4x_5 +0.2264x_4x_6 -0.2702x_5x_6 +0.0040x_1^2 -0.0721x_2^2 -0.0198x_3^2 -0.2076x_4^2 +0.2160x_5^2 -0.0223x_6^2$ |
| Jun | $CY_{t-1}$ | DA6 | DA9 | DA12 | | | $-23.8562 -0.3639x_1 -1.8924x_2 -0.0052x_3 +1.3074x_4 -0.0060x_1x_2 -0.0057x_1x_3 +0.0205x_1x_4 -0.0135x_2x_3 -0.0965x_2x_4 +0.1034x_3x_4 +0.0004x_1^2 +0.0110x_2^2 -0.0171x_3^2 +0.0913x_4^2$ |
| Jul | $CY_{t-1}$ | DA1 | DA3 | DA6 | DA9 | DA12 | $-18.8884 -0.7725x_1 +2.8997x_2 -1.9129x_3 -0.9194x_4 -0.5636x_5 -0.6886x_6 -0.0070x_1x_2 +0.0320x_1x_3 -0.0220x_1x_4 -0.0221x_1x_5 -0.0042x_1x_6 +0.3776x_2x_3 -0.0748x_2x_4 -0.1803x_2x_5 -0.2590x_2x_6 -0.5984x_3x_4 +0.6811x_3x_5 -0.0178x_3x_6 +0.8957x_4x_5 +0.0173x_4x_6 -0.1524x_5x_6 +0.0012x_1^2 -0.1151x_2^2 -0.1006x_3^2 -0.0306x_4^2 -0.7603x_5^2 +0.1200x_6^2$ |
| Aug | $CY_{t-1}$ | DA1 | DA3 | DA6 | DA9 | DA12 | $4.8997 -0.7900x_1 -0.9225x_2 +3.8372x_3 -0.0832x_4 -9.7835x_5 +4.0199x_6 -0.0065x_1x_2 +0.0352x_1x_3 +0.0005x_1x_4 -0.0461x_1x_5 -0.0019x_1x_6 -0.0759x_2x_3 -0.1196x_2x_4 +0.1775x_2x_5 +0.0748x_2x_6 +0.0694x_3x_4 +0.2503x_3x_5 -0.3715x_3x_6 -0.2022x_4x_5 +0.4167x_4x_6 -0.2192x_5x_6$ |
| Sep | $CY_{t-1}$ | DA1 | DA3 | DA6 | DA9 | | $41.4745 -0.5431x_1 -0.0366x_2 -0.9681x_3 +3.6023x_4 -4.3272x_5 -0.0002x_1x_2 +0.0115x_1x_3 -0.0191x_1x_4 +0.0139x_1x_5 -0.0809x_2x_3 +0.0508x_2x_4 +0.0205x_2x_5 +0.4602x_3x_4 -0.5016x_3x_5 +0.3000x_4x_5 +0.0002x_1^2 +0.0172x_2^2 -0.0339x_3^2 -0.3409x_4^2 +0.0831x_5^2$ |
| Oct | $CY_{t-1}$ | DA1 | DA3 | DA6 | | | $-48.806 -0.6966x_1 -0.4241x_2 -1.7664x_3 -3.0097x_4 +0.0040x_1x_2 +0.0053x_1x_3 -0.0175x_1x_4 -0.0038x_2x_3 +0.0111x_2x_4 -0.1443x_3x_4 +0.0008x_1^2 +0.0073x_2^2 +0.0861x_3^2 +0.0558x_4^2$ |
| Nov | $CY_{t-1}$ | DA1 | DA3 | DA6 | | | $47.8316 -0.6925x_1 +0.7765x_2 -2.3671x_3 -2.9813x_4 +0.0043x_1x_2 +0.0011x_1x_3 -0.0066x_1x_4 +0.0797x_2x_3 -0.0306x_2x_4 -0.0144x_3x_4 +0.0004x_1^2 -0.0064x_2^2 -0.0407x_3^2 +0.0200x_4^2$ |
| Dec | $CY_{t-1}$ | DA6 | DA9 | | | | $13.0378 -0.5111x_1 +0.5765x_2 -3.4820x_3 +0.0177x_1x_2 -0.0158x_1x_3 +0.0155x_2x_3 +0.0004x_1^2 -0.0691x_2^2 +0.0343x_3^2$ |










### 4.5 Modelling limitations

The modelling limitations of the presented approach are the following.

(1) To determine drought areas, a threshold value of the Standardised Precipitation Evapotranspiration Index (SPEI) drought index (SPEI ≤ -1) was used. Using just one threshold might lead to over or underestimation of the actual drought impacts over crop yield.

(2) Gridded data of SPEI at spatial resolution (0.5°x0.5°) was used in this study over each region individually. Using such a coarse spatial resolution on different region sizes might not capture the drought area correctly, leading to over or underestimating its magnitude.

(3) The study area has a diverse ecosystem of irrigated and rain-fed land, which may influence the correlation between DA and crop yield more or less.

(4) This study assumes that drought is the only causative factor; however, floods negatively impact crop yield in the region, thus in the total production in the regions. Flood impacts are not considered in the models.

(5) Many other factors might influence rice yield, such as market, technologies, management, etc. In this study, it was assumed that drought plays the prominent role.

(6) Insufficient crop yield data for the ML model building was an issue because the CY time series only had one value for each year.

### 4.6 Crop yield calculation systems

The crop yield calculation is often based on at least one or both types of systems, the one based on ground-field visits and the one based on remote-sensing information. Regarding the temporal scale, those based on ground-field visits are usually issued twice or even four times, as in the case of India, depending on the agricultural calendar. On the other hand, in the case of remote-sensing information, they are usually more continuous, in fortnightly or monthly periods, and aggregated by seasonal periods. The calculations are based on data-driven equations to more complicated models based on crop growth and development. About the spatial scale, ground-field visits-based calculations are generally issued for the different cultivation districts or aggregated by regions and the whole country. In the case of remote-sensing-based calculations, it depends on the spatial resolution of the input data. In theory, the outputs can be scaled down to the district level, although calculations aggregated by district, region, and country are often presented in practice. Although the remote-sensing-based systems have and advance over ground-field visits based method by providing information in the early stages of crop growth, the data required for its execution may not always be available. The ML





approach presented here falls into the second group; therefore, it shares similar limitations on
latency, data availability, and spatial and temporal resolution.

**4.7 On the consideration of other factors, types of drought and indices**

Although many drought indices are initially created to analyse a specific type of drought, it is
also possible to identify other drought types for which indices were created by considering
different aggregation periods. In our study, this was the case. For this reason, we do not
emphasise agricultural drought throughout the manuscript because we are not using only
aggregation periods usually used for agricultural drought analysis. From our correlation
analysis between crop yield and drought areas, we infer that different types of drought (i.e.
meteorological, agricultural, and hydrological) affect the crop yield to varying degrees
throughout the months of the crop period. This level of affectation could be considered to build
the ML models by using the different hydro-meteorological variables or selecting different
aggregation periods of the meteorological variables, as was the case in this research.
Although we have tried to describe how the monthly time series on the calculated drought areas
were matched with the seasonal crop yield data to build our ML approach, some readers may
find the procedure complicated to replicate. If this is the case, we propose two alternatives. One
is to consider an agricultural drought indicator, such as those based on soil moisture. The
second is using a single aggregation period and concentrating on the construction of the ML
model, exploring different types of ML models and modelling strategies.
For the agricultural drought assessment, soil moisture is one of the most suitable variables for
correct monitoring and analysis. The use of soil moisture depends mainly on the availability
and accuracy of this information. We envision using soil-moisture-derived drought indicators
in future studies in similar applications like the one presented here.
Methodologies that consider other factors such as agricultural practices, soil properties and
conditions, among others, are ideal to follow; however, this is not always possible. Our study
presents a methodological alternative for predicting crop yield. There are current approaches
for crop yield calculation in the study area, one based on field visits and another based on
remote-sensing inputs. The main drawbacks and advantages are indicated in the Introduction
Sect. Our methodology complements these two mentioned tools by providing crop yield
prediction that can be compared with the current tools, with the difference that our ML
approach produces results before the harvest (i.e. prediction).
Our research could be extended further. In subsequent studies, we consider that irrigation
practices could be analysed, where the best practices could be identified. Our results indicate



that the increase in drought area is highly correlated with the decrease in crop yield. A more
detailed analysis will make it possible to identify the best agricultural management practices,
identify sub-regions more/less vulnerable to the effects of the different types of drought, and
detect various demands on water resources throughout the different farming systems.
The degree of influence of anthropogenic factors, such as farmer operational practices, and
other factors such as soil conditions, or other natural phenomena such as floods, could be
included in the ML approach. One way to implement the above is as follows. Three or more
types of inputs could be classified: anthropogenic, natural, and different types of combinations.
The variable selection analysis could be carried out for each set of inputs to identify the ones
that primarily drive agricultural production. Subsequently, the ML models could be built
following our proposed approach, i.e. the use of ANN models (or similar models) and
equations.
Weighting the drought areas can be another way to include anthropogenic factors or other
variables. Factors calculated with the additional variables can be used to modify the drought
areas. In this way, the areas would be altered to a greater or lesser extent, increasing or
attenuating the effects of the drought.
Another line that we see much development in the future is the construction of ML models
considering the study area spatially discretised in cells. The availability of spatial data is crucial
in this type of analysis; advances in remote sensing and the different earth monitors developed
in the last decades could facilitate the implementation of this spatially-distributed methodology
using more advanced ML approaches.
Finally, this research can also be extended to analyse the climate change scenarios, either to
elucidate the consequences over crop yield or to find the best crop management practices to
face the predicted problems.

## 654    5 Summary and conclusions

This research introduced a step-by-step ML approach for predicting crop yield (CY) with
drought areas (DAs) as input. The ML approach comprises two components. Each component
employs two types of ML models: polynomial regression (PR) and artificial neural network
(ANN). The goal was to build the ML models (ANN and PR) and use them as an integrated
tool to crop yield prediction. The formulas of the PR models were also provided. The ML
approach was applied in three East India regions.




The following conclusions are drawn from this research.
• Based on the performance of PR and ANN models, results show drought area to be a
suitable variable to predict crop yield.
• The correlation analysis between DA and CY showed high negative correlations in
Odisha (region 3). The correlation gradually decreases in Bihar and Jharkhand (region
1) and West Bengal (region 2). These correlation values can be because West Bengal
has better access to irrigation facilities than Odisha and Bihar & Jharkhand.
• On comparing ANN models and PR models, the ANN were more accurate than PR
models to predict crop yield for all regions. This could have been expected since the
drought–crop relationship is a highly non-linear problem.
• It can be concluded that ANN has a high capability to predict CY in the pre-harvesting
stage with good accuracy, considering the drought indicator used (SPEI), which uses
climate variables such as precipitation and temperature (for evapotranspiration
calculation).
From the analysis and findings of this research, the following recommendations can be
provided for further improvement.
• Sensitivity analysis should be performed to identify the parameters that can impact the
model results. For instance, different spatial resolutions of drought indicator and
different thresholds should be investigated.
• Wet extreme events should be considered, especially in the flood-prone regions such as
the coastal areas of West Bengal (region 2) and Odisha (region 3) and North Bihar
(region 1), where floods also influence crop yield.
• Non-climatic factors such as econometric, fertilisers, and management practices might
be considered because they influence crop yield.
• In order to improve the model accuracy, more input data should be used in further
studies. For CY, this can be estimated by remote sensing techniques on a monthly basis
so that the ML models can be built for this temporal resolution and the spatial coverage
can be better addressed.
• The performance of other ML models has to be investigated, especially committee
(ensemble) methods like random forests or boosting methods. In the case of data at
scales less than monthly, the use of deep learning algorithms (e.g. LSTM networks)
could be recommended to explore.
We envision that this research will improve drought monitoring systems for assessing drought
effects. Since it is currently possible to calculate drought areas within these systems, the direct
application of the prediction of drought effects is possible to integrate by following approaches
such as the one presented or similar.



## Coda and data availability

State-wise crop-yield data was retrieved through the Indian Directorate of Economic and Statistics from the Department of Agriculture (DAC) (http://eands.dacnet.nic.in/). The SPEI data was retrieved from the SPEI Global Drought Monitor (https://spei.csic.es). The code is available upon request from the corresponding author.

## Competing interests

An author is member of the editorial board of journal HESS. The peer-review process was guided by an independent editor, and the authors have also no other competing interests to declare.

## Acknowledgements

VD thanks the Mexican National Council for Science and Technology (CONACYT) and Alianza FiiDEM for the study grand 217776/382365. AAAO was supported by the Orange Knowledge Programme (former NFP) and the World Meteorological Organization (WMO). GACP and VD acknowledge the grand No. 2579 of the Prince Albert II of Monaco Foundation. HvL is supported by the H2020 ANYWHERE project (Grant Agreement No. 700099). DS acknowledges the grant No. 17-77-30006 of the Russian Science Foundation, and the Hydroinformatics research fund of IHE Delft in whose framework some research ideas and components were developed. The study is also a contribution to the UNESCO IHP-VII programme (Euro FRIEND-Water project) and the Panta Rhei Initiative on Drought in the Antropocene of the International Association of Hydrological Sciences (IAHS).

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
