# Peer review of "Spatiotemporal changes of drought area as input for a machine-learning approach for crop yield prediction"

_Hydrology and Earth System Sciences, 2022_

## Author Comment (AC1)

**CC1: 'Comment on hess-2022-252', Xiaofeng Li, 08 Jun 2023**

https://doi.org/10.5194/hess-2022-252-CC1

| # | Comment | Reply |
|---|---------|-------|
| 1 | The manuscript uses spatial range as an input variable and utilizes machine learning algorithms for crop yield prediction. This is an interesting and innovative study. This study can provide a new approach and method for crop yield prediction, while also reducing the dependence of crop models on input data.
  Although this study proposes a novel method, the final results have good accuracy and have been compared and analyzed with site observation data, confirming the reliability of the results. However, I still have some questions about some of the content of the manuscript, and the author still needs to revise it and add some explanations. At the same time, there are some format problems in the manuscript, and there are also some citation format problems in the references. The following are detailed comments and suggestions: | We appreciate your feedback and questions, which we answer below.
The organization and writing of the document have been improved.

We hope this new version is much more understandable and structured. |
|  | **Specific modification suggestions:**
**Data:** |  |
| 2 | • In this study, rice was the research objective and there was a lack of introduction to the characteristics of rice cultivation in the study area. In addition, is the rice in this study area significantly affected by drought? Relevant content should be supplemented. | Thanks, more about the importance of rice in the region has been added. According to the literature, rice yield is hardly impacted by drought in the region. |
| 3 | • Although SPEI has a wide range of applications for drought monitoring, this study should also supplement some literature on this indicator in similar research areas and similar research objectives. | Thanks, references have been added to indicate similar research using this drought indicator. |
| 4 | • Should the author supplement the sources of land use type data? | Reference has been included.
The land use is depicted to illustrate how agricultural the region is. However, this data was not further used in the calculation and results. |
|  |  |  |

| | | **Results and discussions:** | |
|---|---|---|---|
| **5** | ● | From Figure 5, it can be observed that the correlation between yield after trend removal and drought area changes over time, but overall, the correlation coefficient is relatively small. Can this result support subsequent analytical applications? | Although the correlation coefficient is small in some months for each aggregation period, Figure 5 shows the seasonal variation where, in some months, the correlation is high in those months of the crop season.

This correlation coefficient differs for each time aggregation, showing a lag between the time series. |
| **6** | ● | From Figure 7 to Figure 9, it can be found that the root mean square error of simulated yield in the three study areas has very high accuracy. Should the applicability and differences between the two methods be appropriately supplemented? | The results description has been improved. RMSE is shown for each month, but it needs to be noted that models are only suitable for the indicated month and previous months. Although models can still be used after the month, it is preferable to use the most suitable. |
| **7** | ● | In section 4.4, a large number of models are listed. Can the author discuss the universality of these models? In addition to accuracy, the applicability and ease of application of the model are key considerations for its future construction. | This comment is linked to the previous one. |
| **8** | ● | In section 4.5, the threshold of SPEI indicators should be supplemented with relevant basis. | SPEI is an indicator widely used in drought studies, possibly the most used after the Standardized Precipitation Index (SPI), so the methodology and the fundamentals of the thresholds are widely known. We have indicated the reference for those interested in details.
Line 119 |

---

## Author Comment (AC2)

**RC1: 'Comment on hess-2022-252', Anonymous Referee #1, 08 Jan 202**

https://doi.org/10.5194/hess-2022-252-RC1

| # | Comment | Reply |
|---|---------|-------|
| 1 | **General comments**
The manuscript (MS) "Spatiotemporal changes of drought area as input for a machine-learning approach for crop yield prediction" written by Diaz et al., which argued the limitation of dynamic crop model in predicting crop yield and thus introduced machine learning (ML) method for yield forecasting in three main rice growing regions in India (1967-2015). Two ML approaches: polynomial regression (PR) and artificial neural network (ANN) were employed to investigate in separated or combined modes using drought area as single input for grain yield prediction. Since ML comes to practices and being helpful tools and different applications in our life nowadays, especially in agriculture such as yield predication, remote sensing, this study and MS could provide meaningful approaches for yield forecasting as complementary knowledge for other existing approaches, especially in India.

The figure and visual features are informative and easy to follow. English grammar was well-written. The data 1967-2015 was also a strong point for this MS.

However, given some major issues which are listed here
(i) the objectives of the work and MS were not well determined and clearly stated
(ii) structure of MS was not in well-designed and formulated with concrete objectives
(iii) a lot of repetition and redundant information among sections, figures and tables were not followed with the main text
(iv) lack of more detailed discussion of how other work/other approaches (crop models + ML) has been done elsewhere (in the Introduction and discussion)
(v) critical issue via using drought area as input for model without clarification of other factor or drought intensity. | We appreciate your comments made through each section of this manuscript, we recognise your specific and general observations.

We have processed your comments; below you can find how we addressed them in the last version.

We hope this new version is much more understandable and structured. |

| | With these, it could not be accepted as the current MS state. Please see many comments and suggestions in detail below. | |
|---|---|---|
| | | |
| | **Abstract** | |
| 2 | Line 20-28: it is a bit too long for approach description while it is lack of concrete (overall) statistical number for the results | The abstract has been updated. We have three case studies and two sets of ML, so placing numerical results is a challenge because more background is needed for the readers. We preferred to be very specific on the motivation and description of this research, which is the most novel part. However, we did modify the abstract to make it more concise. |
| 3 | Line 26: explicitly mentioned to PR, only two approaches here | Polynomial regression (PR) is mentioned because, in our approach, the logic is to use PR equations as the first step in crop forecasting with the available information. Further, the forecast can be updated with new data but now using the ANN models. We developed the approach inspired by the operational drought monitoring task, where preliminary/estimated data is often available with lower resolution and in an aggregated way. This data serves as input to the PR equations and provides a first forecast. After that, drought areas can be calculated with spatially distributed data. Then, the artificial neural network (ANN) models are used for a more accurate forecast. |
| 4 | Line 33: space after "implement" | Thanks, checked. |
| | **Introduction** | |
| 5 | There is redundant information in the first paragraph (line 38-51) that needs to be rewritten. | These paragraphs have been updated. Lines 38-51 |

| 6 | The MS emphasised the limitation of crop modeling which has been well established in long time in crop yield simulation, yield prediction, and climate change impact assessment as well as understandings crop responses to different abiotic or biotic stresses. Both crop models and ML have uncertainties with regards of spatial-temporal input data when bring into larger scales and long-term application. The comparison between ML and crop model should be further elaborated in the text to convince the reader towards ML? (line 52-59). | We believe this paragraph provides the basis for understanding the study's motivation. The paragraph (lines 54-68) includes references for those who wish to delve into more details. |
|---|---|---|
| 7 | Similarly, the MS focused on spatial extent of drought, and it convinced it as an issue that ML model could cover but there is no detail literature and reference that have been done for that in the MS (line 68). Why it is important? | Thanks. We have updated the paragraph, and now it includes the references. Lines 61-63 |
| 8 | Line 78: what are the specific objectives, about spatial extent impact on grain yield prediction in ML or determine which the best approach of ML are or temporal aggregation effects? Please clearly state | The specific objective is the following: *This research aims to develop an ML approach to calculate seasonal crop yield (CY) with the monthly drought areas (DAs) as input. The ML approach comprises two types of models polynomial regression (PR) and artificial neural network (ANN).* The research's objective is indicated in the Introduction (lines 72-81) We updated the text to indicate the specific objective. (lines 72-81) |
| 9 | Line 89-123: paragraph "Crop yield prediction in India" came to this. This section should be rewritten or merged with above section to make the Introduction more streamline with clear issues and associated objectives. The mentioned information in this section was repeated in section 2 | The Introduction section was updated. The text you referred to was removed from the Introduction. Also, Section "2. Data" was updated to check for any repetition. |
| 10 | Line 99-109: writing need to be improved | Thanks, The Introduction was updated, and this paragraph was moved to Sect. 4.7 |

| 11 | Line 119: which are "other solutions"?

Is there any study using the drought area for yield prediction before? | Other solutions refer to those that do not depend on satellite information directly, for example, the time series of data estimated or gauged by other non-remote sensors, for example, sensors located in the field.

To the best of our knowledge, no other studies have explored drought area in the way we use it for crop yield prediction, i.e. using different drought area ranges as a proxy for drought magnitude. |
|---|---|---|
| | **Materials and Methods** | |
| 12 | Section 2 and 3 need to be reconstructed for more concise and easy reading. It is better to merge in one: like "Materials and methods" with further subheadings. | Thanks, we structured these sections as "2. Data" and "3. Machine Learning methodology" to follow a logical application to build the ML models.
In addition, in practice, the reader can use the Methodology (Sect. 3) with drought area data calculated with any other drought index, not necessarily SPEI. However, we have updated the writing to be more precise and easier to understand. |
| 13 | Line 131-135 is repetition with lines 99-102 | The paragraph was updated to avoid any repetition.
Lines 99-104 |
| 14 | Line 131: accessed when? Also the DAC is not similar to the name in line 95 | DAC (Directorate of Economic and Statistics from the Department of Agriculture) and DESMOA (Directorate of Economics and Statistics, Ministry of Agriculture) are acronyms for different organisations.
On the other hand, the last access to the source has been placed.
Last accessed November 23, 2021. |
| 15 | Line 143, separately for each state? | The state of Bihar suffered a political separation, so the territory was divided into two: Bihar and Jharkhand. Data is now reported separately for each of these states. This is highlighted because the Bihar data, in general, the current values are lower after Bihar's division. Our research used a single time series for the entire original region. |
| 16 | Line 145: it is not clear, it is the spatial aggregation of two states with the average yield? | The paragraph was updated. From 2000, the time series is the sum of the smaller Bihar and the new state Jharkhand. |

| 17 | Figure 1: Why are the color of left and right figures are so different? Same color scale? What is spatial resolution of grid at legend? | Figure 1 was updated to avoid any confusion. The left figure shows only the location of the three regions; no cropland data is depicted. The spatial resolution is 0.5 deg, it was also added in Figure 1. |
|----|---|---|
| 18 | Line 156: there is no reference on the reference list | Thanks, the reference has been added. |
| 19 | Line 160: access when? | The date has been added. Last accessed November 23, 2021 |
| 20 | Line 162-163: this information is really important for the whole MS that do not need to repeat explanations. Please state clearly the aggregation: how to get DI and DA? DA1 is aggregated of what from when to when? And soon DA3, 6, 9, 12 because it is confusing with 12 months or 24 months (line 245, 246). | This section describes the drought indicator data and how drought areas were calculated from the drought indicator.

The monitor provides several temporal resolutions for the drought indicator (called aggregation periods). We focused on the description of the data that was downloaded.

On the other hand, to describe how the areas (DA) were processed, it is first necessary to describe how these areas were calculated. All this is described in detail in section "3.1 Data Preparation".

Moreover, line 245-246 was updated to avoid any confusion (Line 124-128).

Section 3 was also updated to clarify how the drought indicator data was processed (Sect. 3.1.2 Drought areas calculation). |
| 21 | Line 185-203 and section 2.2 was rather replicated. | We do not find replication in these sections. Lines 185-203 explain the procedure to calculate drought areas and Sect. 2.2 describes the drought indicator data that was processed. |

| 22 | It is really important to explain further how to estimate such SPEI, in term of equation, variables and since this is only input for the model. The MS mentioned many times the limitation of different drought types, by explanations further this SPEI could determine or clearly show drought? Which ET approach was used and climatic variables? Information of irrigation (if it is available) should be mentioned and described for all years. | SPEI is an indicator widely used in drought studies, possibly the most used after the Standardized Precipitation Index (SPI), so the methodology is widely known. In the paper, we mention that in the lack of soil moisture data, which has proven to be better for the analysis of agricultural drought, formulation of the SPEI can be followed. We do not compare nor test SPEI as a better indicator for agricultural drought. We provide the reference for the reader interested in how SPEI is calculated. We do not calculate SPEI but use the data from the drought monitor in different aggression periods.

On the other hand, the irrigation data presented in Fig. 5 is shown to facilitate the discussion of the results. The results change among the different basins, which is the degree of irrigation among other possible drivers. No more detailed analysis is done regarding irrigation. This analysis is undoubtedly interesting to carry out in future applications and extensions of the developed approach. |
| --- | --- | --- |

| 23 | Using a single input variable like DA might not be concrete enough for yield prediction and the soundness of approach is rather weak, how about other climatic factors like temperature? How is uncertainties of SPEI at global scales? | Including temperature or other variables is an interesting research that requires further development and is out of this work's scope.

We emphasise that we do not use a simple time series of DA, but the arrangement of different drought areas from several temporal aggregations of the drought indicator; this is an indirect way of considering drought areas of different types of drought (meteorological, agricultural, and hydrological), which occur with a lag between them, from meteorological to hydrological. These drought areas are "intelligently" and "weighted/integrated" to calculate crop yield using the ML models. This way of approaching crop yield prediction is novel as far as we know.

Of course, this approach is subject to improvements, which could go in different directions, from the inclusion/testing of other drought indicators or/and including other variables (not necessarily drought areas/indicators), to building another type of ML models. Also by going from this approach where time series are used to a more fully spatial one, with the help of deep learning. |
|----|----|----|
| 24 | Figure 2 should be right away after line 203 | The figures in the previous manuscript were placed in locations that further reduce the white/empty spaces.
 On this occasion, in response to his comment, the figures have been placed in the manuscript just after being mentioned for the first time. |
| 25 | Line 207 how about pest and diseases, heat stress, ozone? | These are also factors that require further analysis, we mentioned in the text to indicate some examples.
Lines 163-164 |
| 26 | Line 229-237 was repeated somewhere else before, for instance line 160-163 or 199-203 | Sections have been updated.
Lines 183-191 |
| 27 | Section 3.2. it was too long and need to be sharpened due to a lot of repeated information | Section 3.2 has been updated. |
| 28 | Line 280: Table 2 should be mentioned right away. Line 280 to 289 should be in the result and discussion section, i.e. line 457 | We have restructured the section. Table 2 and lines 280-289 have been moved and text has been updated in the Results and Discussion section. |

| 29 | Section 3.3 need to be restructured following subsequence equations | This section shows the four types of PR models that were used. We do not consider that the section should be restructured. A clear distinction is made between PR (Step 3) and ANN models (Step 4, Sect. 3.4), which facilitates constructing both models. |
|---|---|---|
| 30 | Section 3.4 also too long and overlapped with the Introduction. Did the work choose the FFNN? | Thanks, the Section 3.4 has been updated. |
| 31 | Line 346? Is that a common threshold for different objects? Any justification to use this threshold for single input variable model? | Reference has been added. Line 272 |
| 32 | Line 350: is that "period" or whole dataset? | Whole dataset, the text has been updated Line 275. |
| 33 | Section 3.5: mentioning various approaches but which one do you choose and what are criteria that has been used? | Thanks, we now describe the criteria for applying the built models. Lines 280-285 |
| | **Results and Discussion** | |
| 34 | It was too lengthy and repeated information. Substantial improvement in writing is required to make the MS well-structured following the objectives with good discussion and reflections with previous studies | Unfortunately, there are not many works similar to ours. There are examples of using drought indices, but not spatial characteristics of the drought, such as drought area. We have structured the results and discussion section to align it with the methodology section; by doing it in this way, we think the reader can follow the methodology and replicate it. The section has been updated anyway. |
| 35 | Line 362-366: legend does the job. | Text was updated. |
| 36 | Line 368: "theree" -> "three" | Text was updated. |
| 37 | Line 394: the decrease and maximum of what? | Drought area (DA), the text was updated. |
| 38 | Line 394: where is Figure 4? It should be shown directly. | Figures have now been placed immediately after being mentioned for the first time. |
| 39 | Any explanations of the de-trended yield from 2003-2015 of region 1 was much fluctuated as compared to region 2 and 3 in the same period? | Thanks for this observation. In the three regions, the de-trended CY fluctuations are more frequent in some periods than others. For example, in region 1, the fluctuation is more frequent from 2003 to 2015; region 2, from 1967 to 2001; and region 3, it is also more evident from 1967 to 2001. From the three regions, region 1 is the most northerly located. Lines 311-314 |

| 40 | Line 403: why is so much different in three regions although only yield from Kharif was presented? Any studies before? | The correlation results between changes in CY and DAs are as expected. Figure 3 shows how different the magnitude of changes in DAs is in the three regions; Figure 4 shows how changes in CY fluctuate differently over the period, so the correlations are different, but, as noted, the highest correlations between drought area changes and crop yield changes are within the crop season. |
|---|---|---|
| 41 | Line 407: what is SPEI6? | Standardised Precipitation-Evapotranspiration Index (SPEI) for 6 months of time aggregation.
Text was updated.
Line 338 |
| 42 | Line 411-416 about figure 5: peak of what and in which figure? 5a 5b or 5c, please more precise | Peak of correlation coefficient (R).
Lines 341-349 |
| 43 | Figure 5: each point on 5 a, b, and c from how many n sample? Line 440: "rein" -> "rain"
Line 441: data for "2014 or for which years? Or average of which years? This is very important information together with SPEI and DA that should be used to interpret the input data and yield prediction results. | For 49 years of CY, please see section 3.1

Thanks, typos have been corrected.

2014 is the year of the reference.

Unfortunately, no time series of irrigated and rain-fed agriculture were retrieved and processed. However, we used the information depicted in Figure 5d in our result and discussion sections. |
| 44 | Figure 447 (figure 6): ", respectively" Is that correlation coefficiency with significant level of 95% | No significance test was carried out. Figure 6 shows the results of the coefficient correlation calculated for each month.
Results are used to select the variables to build the ML models. |
| 45 | Line 466-470 is redundant since it has mentioned in the material and method. | Thanks, text has been updated. |
| 46 | Section 4.3 too much information was shown in same time, fig. 7, 8, 9 as once but less discussion and comparison with other literature for this section. Is there any study elsewhere has been done? | Unfortunately, there are no similar studies, but still, we discuss our results. |

| 47 | Is there any explanation why both models are less accurate from around 2000-2015 as compared to 1967-2000 for instance for region 1 and region 3? Authors mentioned about the "spatial extent" which was considered in the models. But, this was not well discussed. | Spatial extent refers to the area of drought, which, as shown in the models, is a good proxy for drought intensity. In the period mentioned, more significant fluctuations of CY are presented, although not perfectly, the models manage to capture these fluctuations in CY using the changes of drought areas. |
|---|---|---|
| 48 | Section 4.4. Table 4, 5, 6 could be moved to Supplementary material if this is possible since these has not been discussed much or not informative. Line 539, 547, and 556: "moth" -> "month" | Thanks, we believe Tables 4, 5, and 6 provide readers with a more comprehensive understanding of the PR models.

Thanks, the typos were corrected. |
| 49 | Section 4.5 The limitation was listed but has not been shown through the discussion of results and how they affected to the model performance? Or they has not been clearly discussed and compared with other studies? | These limitations refer specifically to this study. We have grouped all of them in this section to help the reader understand our approach's scope and guide him/her in future applications and developments. |
| 50 | Point 6 (line 580-581) it is not clear. In fact, India could provide 3 sets of yield data per year (three growing seasons). Three sets of yield could correspond to at least three periods of temporal aggregation. Why did the work not take three sets of yield data then having more grain yield data with montly DA? | This is an excellent observation, we have modified the text. We have limited our study to just one crop yield season, the largest one. Future implementation can benefit from the other two field samples data. The text has been updated. Lines 507-508 |
| 51 | Section 4.6: Repetition of Introduction and too general without literature comparison and discussion. | Text has been updated. Now, Sect. 4.7 |
| 52 | Line 596-598: is similar to point 2 Section 4.5 | Text was updated. |
| 53 | Section 4.7 a lot information was mentioned and repeated with the previous section line 4.5 and 4.6 | Text was updated. Now, Sect. 4.8 |

---

## Author Comment (AC3)

**RC2: 'Comment on hess-2022-252', Anonymous Referee #2, 22 Jul 2023**

https://doi.org/10.5194/hess-2022-252-RC2

| # | Comment | Reply |
|---|---------|-------|
| 1 | In this manuscript, the authors employ data-driven techniques to predict rice crop yields in India. The paper's objective is clear; however, the methodology is not rigorously employed, the novelty is limited, and the document's structure could be enhanced. In order to improve the study, the authors could consider the following points: | We appreciate your comments. The document has been updated, improving the structure and writing. We mention in the comments below the novelty of the work and also indicate what is related to the methodology, we hope that in its new form, this new manuscript will be of your approval. |
| 2 | 1. In lines 64-68 you mention that ML techniques have already been tested to predict crop yield but that "the use of spatial characteristics of drought such as its spatial extent has not been fully explored to crop yield prediction". Does this mean that the only conceptual novelty of this work is that it considers a new variable? | We introduce an innovative approach to predict crop yield using spatiotemporal changes in drought areas. Most of the previous work has been focused more on the development of indicators and on the use of multivariate methodologies to improve crop yield prediction. In this research, we proved that changes in drought areas are a good indicator of how drought negatively impacts crop yield. Another novel element is the conceptualisation of the approach, we used two types of ML models: polynomial regression (PR) and artificial neural network (ANN) as integrated tool. Lines 69-81 |
| 3 | 2. The authors write in the Modeling Limitations section that insufficient crop yield data is an issue, however, the last year for which crop yield data is available is 2015, is it possible to increase the dataset? Much more importantly, the basis of data-driven techniques (of which ML algorithms are part) is that a lot of information is available, and the algorithm can learn from the data. If you don't have enough information, how can you justify the application of a ML algorithm? | It is possible to increase the period of the data but we foresee that the conclusions are still valid. The use of ML is justified when using not only a single time series of drought areas but many (see Methods and data). We are using monthly drought areas calculated with various aggregation periods of the drought indicator. Moreover, in drought monitoring, the variable aggregation is often done in different periods to try to monitor different types of droughts. In our research, we used the drought indicator with 3, 6, 9, and 12 months aggregation period. |

| 4 | 3. Some of the plots presented in Figure 7 show a serious problem. Your predictions present a lag of one year (the red curve is shifted one year to the right). This usually indicates that an auto-regressive algorithm (like the one that you are using) is not capable of learning and that the prediction of year t+1 is strongly influenced by the crop yield of year t. | Thanks, we included the change in crop yield of the previous season which considerably improves the prediction compared to not using it. In future research, the best order of the crop yield (i.e. t+2, t+3) can be investigated. |
|---|---|---|
| 5 | 4. Go through the entire document and check English usage and typos. | Thanks, we have checked the entire manuscript. |
| 6 | 5. I suggest that the authors revisit the document and avoid repeating information (unless strictly necessary) and avoid presenting graphs with excessive information. | Thanks, the entire document was updated and restructured to address this comment. |
| 7 | 6. You need to improve the description of your work in the introduction. As it is right now, it is unclear. What do you mean by "the crop yield calculation is clear"? What do you mean by "is not as clear"? What does "The ANN is expected to be used with the final input data" mean? | The logic of this integrated tool is as follows. PR provides the prediction where the crop yield calculation is easy-going to the performer (the end-user) because she/he has access to the equations that have a straightforward interpretation, and calculations can be done with early and preliminary input data. For its part, ANN is used as the most accurate model, although the output calculation is not always easy to follow, as in the case of PR, due to the difficulty of interpreting the structure of the resulting ANN. The ANN model is used with the final and more accurate input data.

We have updated the Introduction section. |
| 8 | 7. Did you evaluate the cross-correlation between input variables? Is it possible that you provide redundant information to the algorithm? | We calculated the correlation between inputs and crop yield and based on the described procedure we selected the variables to build the ML models. As we mentioned, in future applications the exploration of best inputs, models, and scales (spatial and temporal) could be done to improve the approach we introduced in this research. |
| 9 | 8. In the results section you write sentences using terms like "perhaps" and "may". However, the results should be able to prove or reject a hypothesis. I strongly recommend that you avoid that type of sentences in the work | Thanks, we have rewritten the Results and Conclusion section to avoid those terms. |